# Randomized resonant metamaterials for single-sensor identification of elastic vibrations

Tianxi Jiang [1], Chong Li[1], Qingbo He [1✉] & Zhi-Ke Peng[1]

Vibrations carry a wealth of useful physical information in various fields. Identifying the multi-source vibration information generally requires a large number of sensors and complex hardware. Compressive sensing has been shown to be able to bypass the traditional sensing requirements by encoding spatial physical fields, but how to encode vibration information remains unexplored. Here we propose a randomized resonant metamaterial with randomly coupled local resonators for single-sensor compressed identification of elastic vibrations. The disordered effective masses of local resonators lead to highly uncorrelated vibration transmissions, and the spatial vibration information can thus be physically encoded. We demonstrate that the spatial vibration information can be reconstructed via a compressive sensing framework, and this metamaterial can be reconfigured while maintaining desirable performance. This randomized resonant metamaterial presents a new perspective for single-sensor vibration sensing via vibration transmission encoding, and potentially offers an approach to simpler sensing devices for many other physical information.

[1] State Key Laboratory of Mechanical System and Vibration, Shanghai Jiao Tong University, 200240 Shanghai, People's Republic of China.
✉email: qbhe@sjtu.edu.cn

Vibrations carry a wealth of physical information in the natural world, which is not only important for perceiving the surrounding environments[1], but also useful in various fields, such as health care monitoring[2,3], earthquake detection[4], smart devices[5], and Internet of Things[6]. Before being picked up by sensors, vibrations are filtered and mixed during propagation. Identifying the information carried by vibrations has great significance[7,8]. In most complex cases, directly measuring vibrations from multiple sources is impossible. The inverse methods are the main approaches to vibration identification by measuring the related quantities[9].

One of the inverse methods is based on the filtering property of unintentionally designed vibration transfer paths to quantify which sources and paths contribute the most to the vibration issues[10]. Other methods, such as blind source separation and array signal processing techniques, can reconstruct vibration information from mixed signals when assuming some source characteristics (e.g., statistical independence and narrowband)[11,12]. In these methods, the accuracy of vibration identification relies on the number and placements of sensors, which requires complex data acquisition systems and control circuits, and results in high power consumption.

Compressive sensing has been demonstrated to be an effective approach to circumvent the traditional sensing requirements[13]. One of the most impressive advances in compressive sensing is the design of the single-pixel camera[14], which has inspired many studies in both the electromagnetic and acoustic fields[15–20]. In these studies, compressive sensing is combined with spatially encoded structures such as random scattering masks and meta-materials. Metamaterials are a broad family of artificially structured materials with unusual effective properties and functionalities[21–25]. Flexible manipulations of electromagnetic, acoustic, and elastic waves can be achieved, such as cloaking[26–28], beaming[29,30], diffusing[31], illusion[32], and hologram[33]. Fascinating applications, such as high-speed analog computing[34,35], ultra-sensitive detection[36–38], and efficient waveguiding[39] have been demonstrated. In the field of acoustics, a properly designed metamaterial achieved encoding of acoustic waves in the spatial and frequency domains, which can be considered as a physical implementation of the measurement matrix in compressive sensing[17,20]. Unlike advances in the field of acoustics, spatial vibration encoding and identification with metamaterials remains unexplored. The core issue is how to design an eligible meta-material with uncorrelated transmissions to encode spatial vibration information.

In this work, we propose a randomized resonant metamaterial with randomly coupled local resonators for single-sensor identification of elastic vibrations. This metamaterial is designed by developing the theory of randomly coupled resonator dynamics. The metamaterial is proved to be capable of producing highly uncorrelated transmissions for different spatial vibrations due to the disordered coupling of random effective masses. Based on encoding spatial vibration information by uncorrelated transmissions, a compressive sensing framework can be used to experimentally identify various vibration events with only a single sensor. We further demonstrate that the designed metamaterial can be used as a new type of human-machine interface. Our study not only is applicable to areas such as smart devices and Internet of Things, but also provides exciting perspectives for designing simpler vibration sensing devices and other physical sensing systems.

## Results

### Randomly coupled resonator system for vibration encoding.
To achieve the highly uncorrelated transmissions for spatial vibration information encoding, we propose a concept of randomly coupled resonator dynamics and develop the corresponding system. The proposed system consists of multiple different coupling networks. Each coupling network is composed of $N$ local resonators connected by springs $k_0$ and dampers $c_0$ (Fig. 1a). $k_n$ and $c_n$ are the stiffness and the damping coefficients of the $n$th resonator, respectively. $x_n$ and $y_n$ are displacements of matrix $M$ and mass $m$. We derive the randomly coupled resonator dynamics (see Methods for details). In the dynamics, the local resonator is simplified to be an effective mass as below

$$m_n^{\text{eff}} = M + \frac{k_n^{\text{d}} m}{k_n^{\text{d}} - \omega^2 m},\tag{1}$$

where $k_n^{\text{d}} = k_n + i\omega c_n, k_n = m\omega_n^2$ and $\omega_n$ is the resonant frequency of the $n$th local resonator[40]. The effective mass is frequency-dependent and can be negative near the resonant frequency due to the anti-resonance effect, which corresponds to the attenuation of elastic waves (Supplementary Fig. 1).

In the coupling network, the effective masses of all local resonators can be the same or different. For convenience, we refer to the coupling network with the same resonators as an ordinary coupling network (OCN) and that with random resonators as a randomized coupling network (RCN). By using the finite element method (FEM) and the analytical method, the transmissions of an OCN (with a local resonance frequency of 420 Hz) and a RCN can be obtained (Fig. 1b) (see Methods for details of numerical simulation and analytical derivation). The analytical results agree well with the numerical results. For the OCN, the transmission has a wide attenuation region (gray shaded region in Fig. 1b) due to the presence of the local resonance band gap[41]. If the OCN is used for sensing, the vibration information in the band gap will be lost, which is disadvantageous for vibration encoding and identification. In contrast, the RCN can randomly encode vibration information in the frequency domain, because the randomized effective masses lead to random distributions of the amplification and attenuation bands. This randomized encoding property is useful for constructing a desirable measurement matrix for the compressive sensing.

Figure 1c shows an entire randomly coupled resonator system by connecting six RCNs and a center module (without local resonators) for spatial vibration encoding. In each RCN, the spatial distribution is disordered for the effective masses of resonators. Vibrations generated from multiple sources propagate from the boundaries to the center module. The transmission of the coupling system can be obtained by measuring the vibration signals of the center mass $M_0$. Figure 1d shows the transmissions of the coupling system when excitations are applied at two different locations (see Supplementary Figure 4a for transmissions corresponding to the other locations). It can be seen that substantial differences exist between the two transmissions. To evaluate the uncorrelation of the transmissions, we calculate the correlation coefficients and the correlation matrix $\mathbf{C}_\mu$ (see Supplementary Note 1). The absolute values of elements in $\mathbf{C}_\mu$ are visualized as shown in Supplementary Fig. 4b. The histogram of the absolute cross-correlation coefficients show that they narrowly distribute around zero (Supplementary Fig. 4c). We choose the average of the absolute cross-correlation coefficients $\mu_{\text{Ave}}$ as the metric of the uncorrelation. Here, the $\mu_{\text{Ave}}$ is approximately equal to 0.15, indicating that the transmissions are desirably uncorrelated.

To explain the relationship between the uncorrelation and the disordered effective masses, we analyze the vibration modes of the entire coupling system. Figure 1e shows the vibration modes obtained by, respectively, applying excitations to six locations at 600 Hz. It can be seen that the displacement fields are obviously

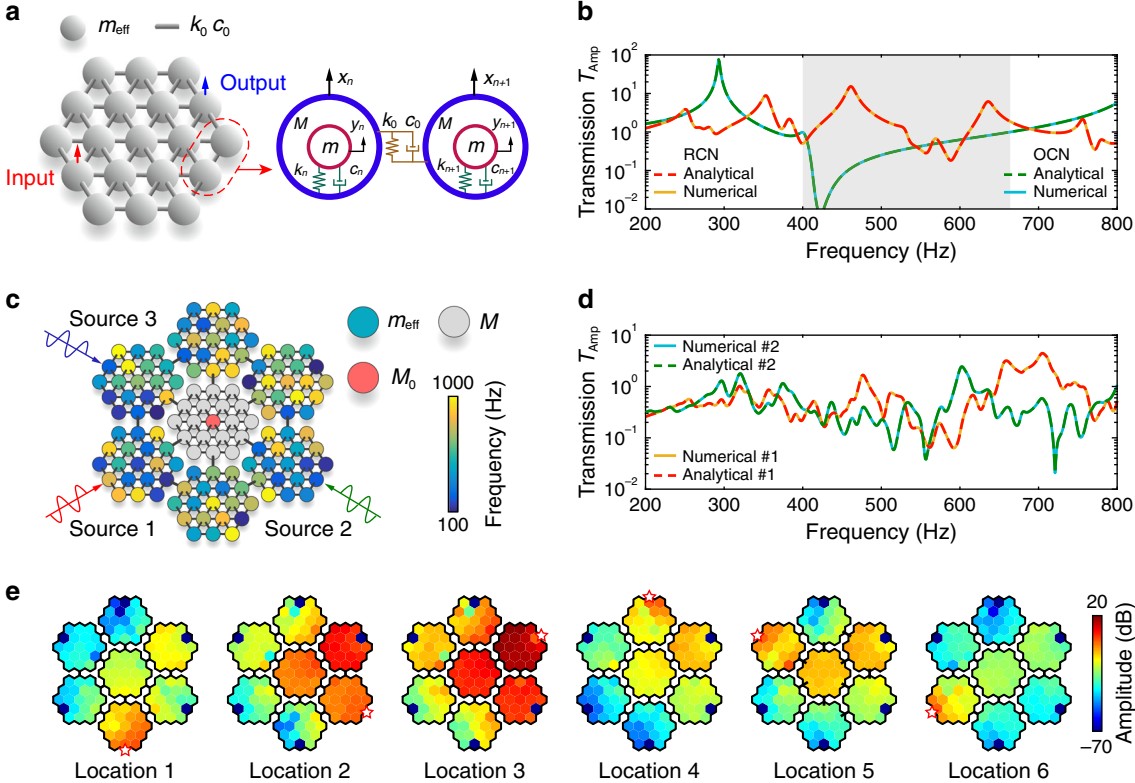

**Fig. 1 Randomly coupled resonator system for spatial vibration encoding. a** Schematic of a simplified coupling network. **b** Transmissions of the ordinary coupling network (OCN) and the randomized coupling network (RCN) calculated by analytical and numerical methods. **c** The entire coupling system composed of six different RCNs for vibration identification. The colors of the resonators denote the resonant frequencies. **d** Transmissions of the coupling system excited at two terminals respectively. **e** Vibration modes of the coupling system excited at different terminals at 600 Hz.

different, which reflects the complexity of the elastic vibration propagation in the coupling system. From the randomly coupled resonator dynamics, we can find that this complex vibration propagation is determined by the effective mass matrix and the coupling matrix of the coupling system. These two matrices reflect the disordered coupling relationship of random effective masses. The disordered coupling of the effective masses leads to the high uncorrelation of vibration transmissions, which is the physical basis for effective spatial vibration encoding.

We also investigate the effects of the model parameters on the vibration transmission property of the randomly coupled resonator system (see Supplementary Note 2). The relationship between the $\mu_{Ave}$ and the parameters, including $c_n$, $c_0$, $k_0$, and the number of the local resonators, can be found in Supplementary Figs. 5 and 6. The results show that the stiffness of the matrix $k_0$, the damping of the resonators $c_n$, and the number of the resonators have a great influence on the uncorrelation of the vibration transmission, which provides a guidance for the optimization model design.

**Design of the randomized resonant metamaterial.** The proposed randomly coupled resonator system with disordered effective masses provides the physical basis for designing an actual metamaterial system, which is feasible for uncorrelated vibration encoding. Here, we choose the spiral-based resonators to realize the metamaterial because their physical properties can be easily tuned by changing the structural parameters[30,42–44]. The designed metamaterial consists of six different supercells surrounding a hexagon plate (Fig. 2a). Each supercell is carefully designed by randomly coupled resonators so that the transmissions of the metamaterial system are sensitive to the spatial

locations. A single acceleration sensor is attached to the center plate to pick up the mixed modulated vibration signals generated from multiple sources. In this way, the sensing system can be expressed as $\mathbf{y} = \mathbf{Mx}$, where $\mathbf{y}$ is the vector form of the measured data from the single sensor (i.e., observation vector), $\mathbf{x}$ is the object vector containing the information of sources, and $\mathbf{M}$ is the measurement matrix determined by the encoding property of the metamaterial and the pre-knowledge of vibration excitations.

The supercell of the metamaterial system is composed of 19 unit cells with random geometric parameters on spiral beams (Fig. 2b). Each unit cell includes two Archimedean spiral beams and a center mass assembly consisting of a bolt and a nut. The parametric equation that determines the spiral beams is given by $x(s) = (\alpha + \beta s + w)\cos(s)$, $y(s) = (\alpha + \beta s + w)\sin(s)$, where $\alpha = 1.2$ mm, $\beta = 0.68$ mm rad$^{-1}$, $w = 1.4$ mm, $s \in [\pi, \theta]$, and $\theta$ is a random variable from $2.6\pi$ to $4.6\pi$. The lattice constant $A_0$ and the height $h$ of the unit cell are 25 mm and 4 mm, respectively. The spiral beams and matrix of the unit cell are made of poly-lactic acid (PLA) with a Young's modulus $E_{PLA} = 3.8$ GPa, shear modulus $G_{PLA} = 1.3$ GPa, and mass density $\rho_{PLA} = 1250$ kg m$^{-3}$. The center mass of the unit cell is 2.52 g. Figure 2c shows the dispersion curves calculated along the Γ-M direction by applying Bloch boundary conditions to the unit cell with $\theta = 2.952\pi$. We also calculate the frequency-dependent effective mass density[45]. The out-of-plane band gap between 304.0 Hz and 493.7 Hz is consistent with the frequency band of the negative effective mass density. The out-of-plane local resonance modes at the edges of the band gap, denoted as A1 and A2, are responsible for forming the band gap. The starting frequency of the band gap can be tuned by varying the angle $\theta$ of the spiral beams, as shown in Fig. 2d. Therefore, the geometric parameter of the unit cell can be derived from the fitting function when the local resonance

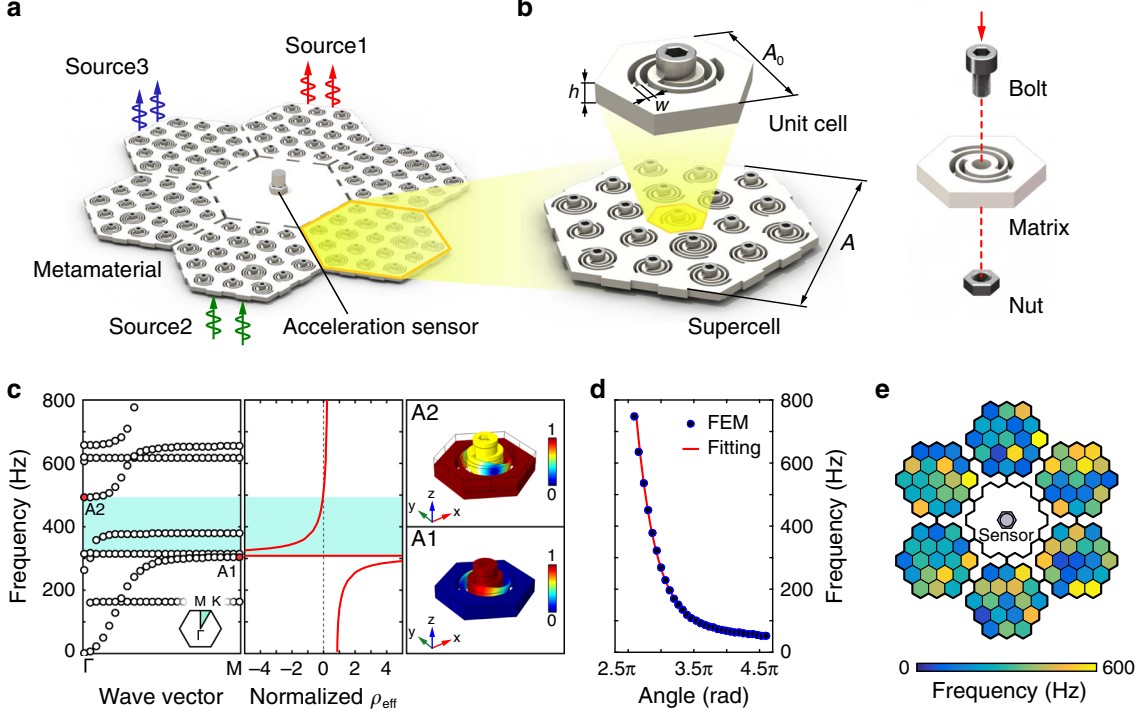

**Fig. 2 Randomized resonant metamaterial system for multi-source vibration identification. a** Schematic of the randomized resonant metamaterial system. **b** The supercell composed of 19 unit cells with different geometric parameters of spiral beams. (**c**) The dispersion curves of the unit cell with $\theta =$ $2.952\pi$. The out-of-plane band gap is from 304.0 Hz to 493.7 Hz. The frequency range of the negative effective mass density is consistent with the out-of-plane band gap. The vibration modes at edges of the band gap are marked by A1 and A2. **d** The relationship between the local resonance frequency and the angle $\theta$. The results calculated by the finite element method (FEM) are marked by circles, and the fitting curve is marked by the red solid line. **e** Spatial distribution of local resonance frequencies of the entire metamaterial.

frequency is given. Because the unit cells are completely different from each other in the supercell, a randomized modulation of vibrations can be achieved over a wide frequency range. Figure 2e shows the distribution of local resonance frequencies of the entire metamaterial system, where a color block corresponds to a unit cell (see Supplementary Table 3 for the parameters of unit cells). This disordered design ensures that the metamaterial system has highly uncorrelated transmissions when vibrations are excited at different locations. The highly uncorrelated transmissions can encode the information of multi-source vibrations to construct the measurement matrix **M**. Once **M** and **y** are determined, the object vector **x** can be reconstructed by solving the inverse problem using compressive sensing theory to realize vibration identification.

To investigate the encoding performance of the metamaterial system, we conduct experiments on a sample of the metamaterial as shown in Fig. 3a (see Methods). Six speakers generating vibration signals are connected to the metamaterial system at different locations. Spatial transmissions can be obtained by using the experimental modal analysis method (Fig. 3b). The correlation of the transmissions (Fig. 3c) is calculated by using the same method in Supplementary Note 1, and the average cross-correlation coefficient $\mu_{Ave}$ is achieved as 0.15. The results demonstrate that the transmission property of the metamaterial system has desirable uncorrelation, which is beneficial for constructing an ideal measurement matrix satisfying compressive sensing theory. Figure 3d specifically presents the complex vibration velocity field distributions at 695.3 Hz, which are experimentally measured by applying excitations to different locations. The results visually show the high complexity of the elastic vibration propagation in the metamaterial, which confirms the effective spatial vibration encoding ability of the designed metamaterial.

**Single-sensor vibration identification**. To verify the performance of the metamaterial system for vibration identification, we experimentally conduct the identification tasks for multi-source vibration information. Twenty different testing signals with normalized energy are collected in a signal set (see Methods for the construction of the testing signals). Because the identification needs a priori knowledge of the measurement matrix, a calibration process is experimentally performed in advance to construct the measurement matrix **M** by calculating the spectra of the measured signals (see Methods). After that, the testing signals are simultaneously emitted from two or three randomly selected sources out of six possible locations. This testing process is performed 40 times. The signals played each time are randomly selected from the signal set. The data measured by the single acceleration sensor form the observation vector **y**. A preprocessing is conducted to compress the dimensionality of the measurement matrix while maintaining the reconstruction accuracy and further improving the robustness against noises by using principal component analysis (see Methods). Then, the object vector **x**, which contains the information of vibration source locations and signal indices, can be reconstructed by using the two-step iterative shrinkage/thresholding (TWIST) algorithm[46]. Specifically, we take an example to demonstrate the entire identification process in detail in Supplementary Note 3 and Supplementary Fig. 8.

Figure 4a shows a reconstruction result with three activated sources, where the blocks in the darkest color represent the most likely locations and signal indices. We denote this configuration of the metamaterial by Config. 1. The reconstruction results agree well with the truth (see Supplementary Fig. 9 for more results). To succinctly characterize the total reconstruction results of the 40 experiments, the reconstructed **x** is divided into six segments. The

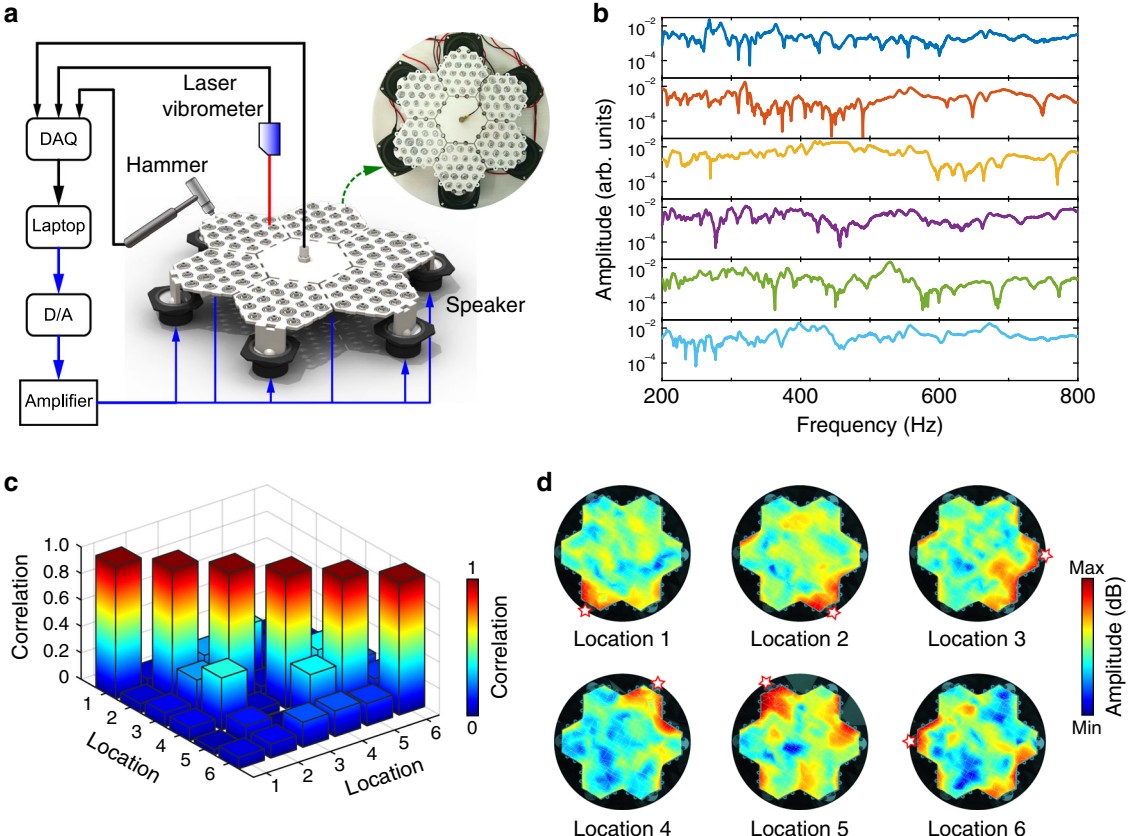

**Fig. 3 Spatial vibration encoding property of the metamaterial system. a** Experimental setup to test the metamaterial system. **b** Transmissions when vibrations are excited at six different locations. **c** Correlation of the transmissions. **d** Vibration modes of the metamaterial excited at different locations at 695.3 Hz.

maximum elements in each segment are chosen and plotted together. The locations of the vibration sources can thus be clearly reflected if they are correctly identified (Supplementary Fig. 10). Here, we define it as a correct identification when the highest reconstructed strength of the activated source is above 0.5 (whereas it is below 0.25 for an unactuated source) and matches the actual locations and signal indices. For this task, the recognition ratio is 96.7%. The results demonstrate that the metamaterial sensing system can accurately identify multi-source vibration information from the single-sensor measurement, which is of great significance for vibration sensing.

In addition to the Config. 1 in Fig. 4a, the metamaterial system can be reconfigured into other configurations according to the practical needs. A windmill-shaped metamaterial system (denoted by Config. 2) is assembled as shown in Fig. 4b. Vibrations are excited at different positions of the windmill blades. A similar recognition task is performed, and the reconstruction result is shown in Fig. 4b. The total recognition ratio of the 40 experiments is 97.9% (Supplementary Fig. 11). Figure 4c shows the configuration (Config. 3) and the reconstruction result for the case where the metamaterial is embedded in a large acrylic plate. Unlike the above configurations, excitations are not directly applied to the metamaterial. The metamaterial system still has a desirable performance, with a recognition ratio of 97.1% (Supplementary Fig. 12a). Furthermore, we investigate the noise robustness of this metamaterial sensing system. Testing signals are excited at two locations, whereas Gaussian noises (SNR = 0 dB) are excited at another location (Fig. 4d). The reconstructed result shows that the vibration information can still be correctly identified under noise interferences. The recognition ratio of the entire recognition task is 95.0% (Supplementary Fig. 12b). The

above results demonstrate that the reconfigurable property of the metamaterial extends the application ranges of the metamaterial system (e.g., in quadrotor drones and airplane wings) while ensuring the accuracy of vibration identification.

The proposed metamaterial system can also be used to identify impacts as shown in Fig. 4e. Different from the testing signals in the signal set, impacts are sparse in both time and spatial domains. The identification of impacts relies only on the intrinsic property of the metamaterial system. This ensures that various impact events can be identified once the impulse responses of the metamaterial system are calibrated (see Supplementary Note 4 for construction of the measurement matrix for impact identification). When impact events occur, the TWIST algorithm can be used to reconstruct the impact signals from the measured signals. Figure 4f shows the reconstruction results of an impact event with two consecutive impulses applied to Location 1. The occurrence location and time of the reconstructed impact are in good agreement with the actual ones. Figure 4g shows the successful identification of another impact event, with four impulses applied to Location 5. The results above verify the feasibility of the proposed metamaterial system for impact identification.

To demonstrate the broad application prospects of our approach, we use the proposed metamaterial system to track the trajectory of multiple vibration events. Here, eight probes (denoted by "A" to "H") are fixed on a plate where the metamaterial is embedded as shown in Fig. 5a. We tap the probes with the finger in sequence along the illustrated trajectory "A-E-F-B-H-D-C-G-A", and the vibration signals are measured as shown in Fig. 5a by the single sensor. We can clearly see the tracked trajectory according to the identification of vibration events (see the details in Supplementary Note 5). Figure 5b shows

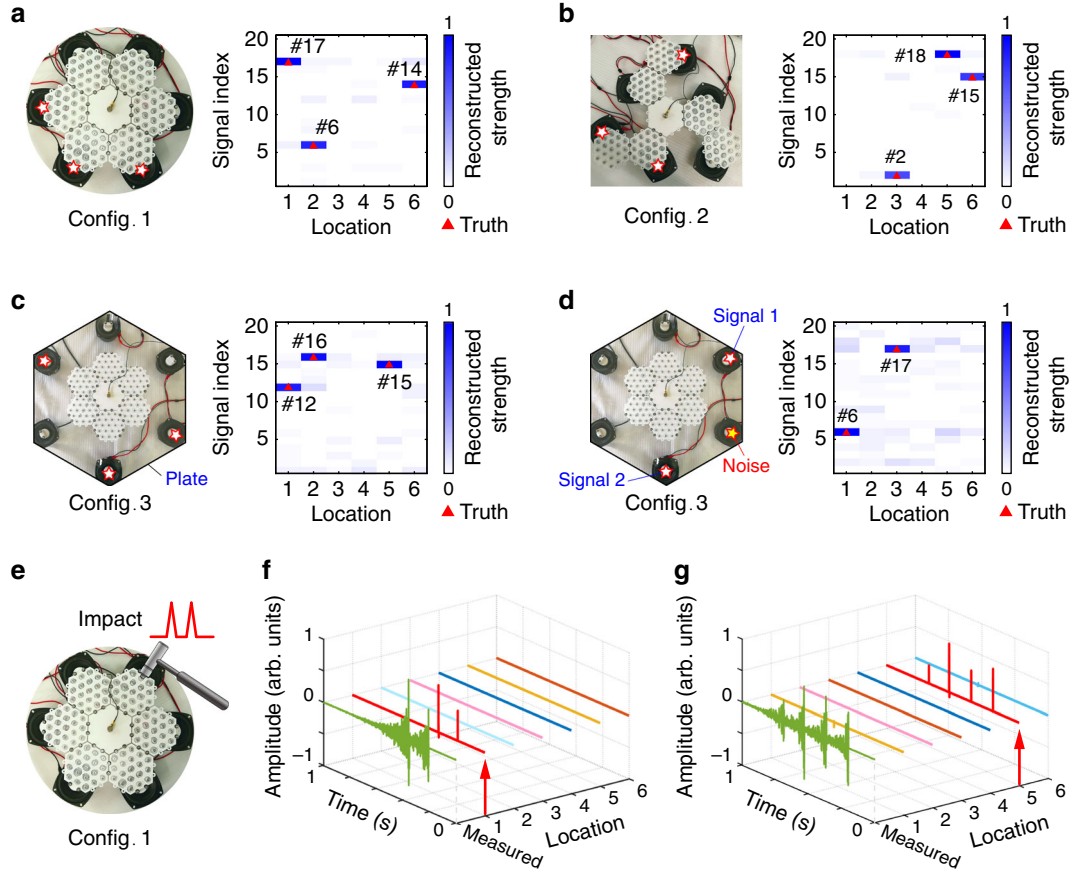

**Fig. 4 Vibration identification of the metamaterial system. a** Experimental setup of Config. 1 and a reconstruction result with three activated vibration sources. **b** Experimental setup and reconstruction results of Config. 2. **c** Experimental setup and reconstruction results of Config. 3. **d** Vibration identification of Config. 3 under noise interferences. **e** Experimental setup for impact identification. **f** Reconstruction result with two consecutive impulses applied to Location 1. **g** Reconstruction result with four consecutive impulses applied to Location 5.

the reconstructed results of each vibration event as well as the trajectory pattern, where the occurrence locations and time of events agree well with the truth (see Supplementary Fig. 13 for an example). This trajectory tracking process is dynamically visualized in Supplementary Movie 1. We also study the tracking process of the trajectories "SJTU" (i.e., the abbreviation of Shanghai Jiao Tong University). The tracked trajectories are well exhibited in Fig. 5c (see Supplementary Fig. 14 for the details). Figure 5d shows the successfully tracked trajectory "Vase" with 12 probes. The experimental details and reconstruction results are presented in Supplementary Fig. 15 and Supplementary Movie 2. The results above demonstrate the time-dependent space coding ability of the proposed metamaterial system, which can create a new type of human-machine interface for instruction, communication, and encryption without complex hardware and high power consumption. Moreover, it also has potential application prospects in fields such as robot tactile sensing and collision tracking.

## Discussion

Our proposed design provides an effective strategy for the spatial encoding and single-sensor identification of elastic vibrations, which can not only reduce the complexity of traditional sensing approaches, but also flexibly meet the practical needs. The proposed randomly coupled resonator system with disordered effective masses provides the theoretical basis for designing the randomized resonant metamaterial and is easy to be extended.

The theoretical concept is a dynamics method that incorporates physical mechanism. The physical mechanism of spatial vibration encoding is that the disordered coupling of effective masses leads to uncorrelated vibration transmissions. In addition, the vibration transmission property of the metamaterial is the synergy of the resonance and anti-resonance of the local resonators. Vibration signals can thus be enhanced to some extent in certain frequency bands, which is superior to the band-stop filtering mechanism[17] and benefit the vibration identification. The designed metamaterial system achieves the identification of multiple vibration events with high recognition ratios, and has proved to be robust to noise interference. Furthermore, we have demonstrated that the designed metamaterial system can be used for trajectory tracking, which has potential application prospects in various fields.

Although the proposed metamaterial system is a prototype, it provides an attractive avenue for simpler vibration sensing, and it can be further improved in the following aspects. The metamaterial can be assembled into more configurations to identify potential spatial vibrations according to practical needs, e.g., a 3D structure configuration[47] to identify omnidirectional elastic vibrations. The operational frequency ranges can be adjusted by changing the structural parameters of unit cells. The size of the system can be further reduced with the help of advanced manufacturing techniques and microelectronics, so that it can be flexibly integrated into numerous smart devices. In addition, the field theory can be introduced to improve the analytical model for further optimization design and performance prediction of the device[28,48].

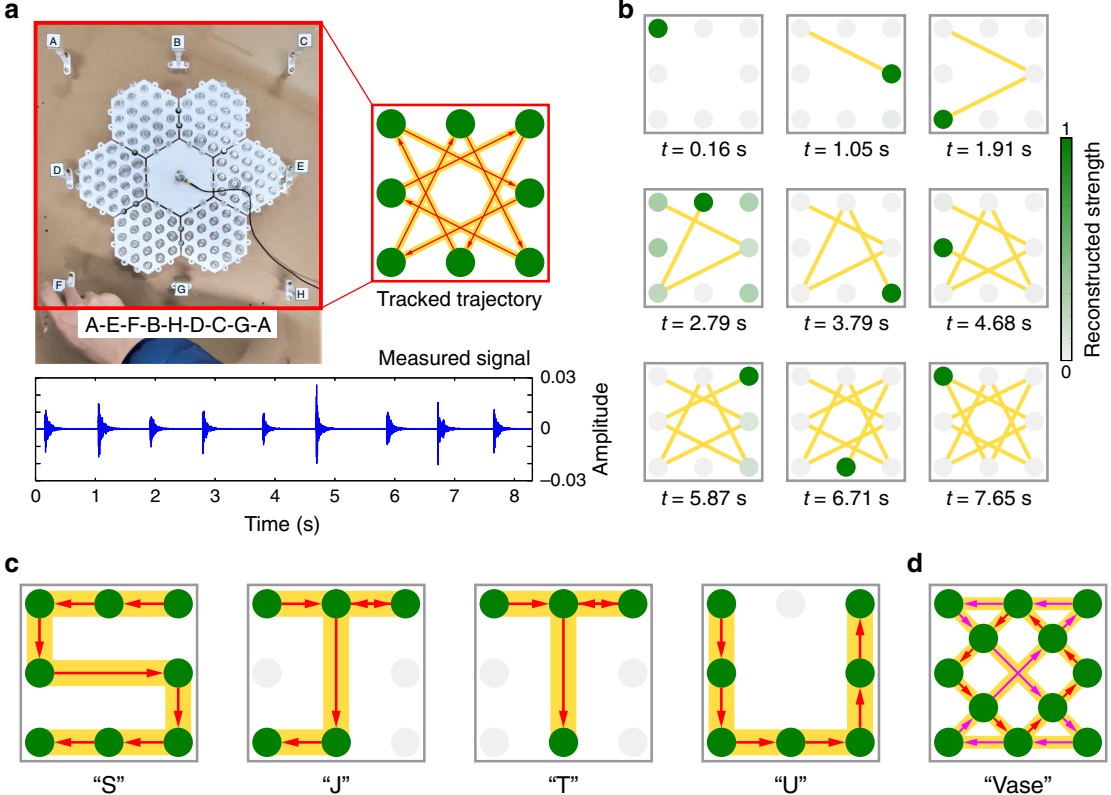

**Fig. 5 Trajectory tracking of vibration events. a** Experimental setup for trajectory tracking with eight probes. The probes are tapped along the trajectory "A-E-F-B-H-D-C-G-A", and the vibration signals are measured by the single sensor. The trajectory is successfully tracked according to the identification of vibration events. **b** Reconstruction results of vibration events. The dynamic process can be found in Supplementary Movie 1. **c** The tracked trajectories of "SJTU". **d** The tracked trajectory "Vase" with 12 probes. The dynamic process can be found in Supplementary Movie 2.

However, there are also limitations for the proposed approach. First, although the proposed metamaterial system has achieved an effective spatial vibration encoding, the randomized design strategy is not the optimal one that can maximize the uncorrelation while minimizing the number of unit cells. This limitation can be overcome by using topological optimization[34,49] or machine learning methods[50] to achieve the on-demand design of the disordered structures in the further work. Moreover, the proposed approach needs the pre-experimental calibrations because the actual sample in experiments might not match the designed one due to the unavoidable manufacturing tolerance. The vibration identification is partly dependent on some pre-knowledge of vibration excitations, which is another reason for pre-experimental calibrations. This issue is hoped to be solved by utilizing artificial intelligence algorithms based on the computational and experimental data[51,52].

Overall, we have demonstrated a randomized resonant metamaterial with disordered effective masses for single-sensor vibration identification. This work opens up avenues of vibration transmission encoding with metamaterials. We envision that the proposed metamaterial can integrate with numerous intelligent devices, platforms, and structures (e.g., wearable devices, quadrotor drones, and airplane wings) so that it can be used in broad fields such as human-machine interaction, health care monitoring, industrial field detection, information processing and communication. Moreover, we believe that the proposed metamaterial design strategy can provide the basis for designing various types of simper sensing devices for vibration and many other physical information.

## Methods

**Numerical simulations**. Numerical simulations are conducted to analyze the randomly coupled resonator system by using the lumped mechanical system interface in the multibody dynamics module of COMSOL Multiphysics Version 5.4. Without loss of generality, the parameters of the randomly coupled resonator system are set as $M = 2 \times 10^{-3}$ kg, $m = 3 \times 10^{-3}$ kg, $k_0 = 1.9 \times 10^5$ N m$^{-1}$, $c_0 = 0.01$ N s m$^{-1}$, $c_n = 0.25$ N s m$^{-1}$, and $N = 19$. The local resonance frequency $f_n$ that determines $k_n$ can be found in Supplementary Table 1. The node configuration of the resonator is provided in Supplementary Fig. 2. Here, nodes "1", "2", and "3" are the internal nodes connecting the masses, springs, and dampers of the resonator. Node "a" is the input node, and nodes "b1", "b2", and "b3" are the output nodes of the subsystem. The coupling network is constructed by connecting multiple subsystems as shown in Supplementary Fig. 3a. The node connection is provided in Supplementary Table 2. Then, a unit excitation is applied to the input terminal of the coupling network. The transmission can be obtained by picking up the signal from the output terminal. The node connection of the entire coupling system is similar to that of the coupling network, and the boundary conditions are shown in Supplementary Fig. 3b. By changing the input nodes, vibration transmissions and modes of the randomly coupled resonator system can be obtained.

**Derivation of the randomly coupled resonator dynamics**. The dynamical equation of the $n$th resonator can be expressed as

$$\begin{cases} M\ddot{x}_n + c_0\left(\sum_{i=1}^{N}\psi_{ni}\dot{x}_i\right) + k_0\left(\sum_{i=1}^{N}\psi_{ni}x_i\right) - c_n(\dot{y}_n - \dot{x}_n) - k_n(y_n - x_n) = F_n, \\ m\ddot{y}_n + c_n(\dot{y}_n - \dot{x}_n) + k_n(y_n - x_n) = 0, \end{cases}$$

(2)

where

$$\psi_{ni} = \begin{cases} -1, & i \text{ couples with } n, \\ \phi_n, & i = n, \\ 0, & \text{others}, \end{cases}$$

(3)

$\phi_n$ is equal to the number of the resonators coupling with the $n$th resonator. Equation (2) can be derived as

$$\left( M + \frac{k_n^d m}{k_n^d - \omega^2 m} \right) \ddot{x}_n + (k_0 + i\omega c_0) \left( \sum_{i=1}^{N} \psi_{ni} x_i \right) = F_n. \qquad (4)$$

The expression of the effective mass (Eq. (1)) can thus be obtained. From Eq. (4), we can express the dynamical equation of the $k$th coupling network as a matrix-vector form of

$$\mathbf{M}_k^{\text{eff}} \ddot{\mathbf{X}}_k + \mathbf{K}_k^c \mathbf{X}_k = \mathbf{F}_k, \qquad (5)$$

where $\mathbf{M}_k^{\text{eff}} = \text{diag}(m_1^{\text{eff}}, m_2^{\text{eff}}, \ldots, m_N^{\text{eff}})$ is a diagonal matrix composed of effective masses given by Eq. (1), and $N$ is the number of the resonators. The displacement vector of the resonators has the form of $\mathbf{X}_k = [x_1, x_2, \ldots, x_N]_k^T = \mathbf{A}_k e^{i\omega t}$, where $\mathbf{A}_k$ is the amplitude vector. The stiffness-damping matrix $\mathbf{K}_k^c$ can be expressed as

$$\mathbf{K}_k^c = (k_0 + i\omega c_0)\mathbf{\Psi}_k, \qquad (6)$$

where $\mathbf{\Psi}_k$ with elements $\psi_{ni}$ is a $N \times N$ coupling matrix reflecting the coupling relationship of the resonators (see Source Data file for the details). The time-harmonic excitation applied to the resonators is $\mathbf{F}_k = [F_1, F_2, \ldots, F_N]_k^T = \mathbf{F}_{k0} e^{i\omega t}$. Thus, the displacement amplitude vector $\mathbf{A}_k$ can be calculated by

$$\mathbf{A}_k = \left[ \mathbf{K}_k^c - \omega^2 \mathbf{M}_k^{\text{eff}} \right]^{-1} \mathbf{F}_{k0}. \qquad (7)$$

Therefore, the transmission of the coupling network can be expressed as

$$T_{\text{Amp}} = |x_{\text{out}}|/|x_{\text{in}}|, \qquad (8)$$

where $|x_{\text{out}}|$ and $|x_{\text{in}}|$ are the elements in $\mathbf{A}_k$ that correspond to the output and the input of the coupling network.

The transmission of the entire coupling system can also be derived in a similar way. The dynamical equation of the coupling system is given by

$$\mathbf{M}_E^{\text{eff}} \ddot{\mathbf{X}}_E + \mathbf{K}_E^c \mathbf{X}_E = \mathbf{F}_E. \qquad (9)$$

Here, $\mathbf{M}_E^{\text{eff}} = \text{diag}(\mathbf{M}_1^{\text{eff}}, \mathbf{M}_2^{\text{eff}}, \ldots, \mathbf{M}_6^{\text{eff}}, \mathbf{M}_0)$, where $\mathbf{M}_0 = \text{diag}(M, \ldots, M)$ is the mass matrix of the center module of the system. $\mathbf{X}_E = \mathbf{A}_E e^{i\omega t}$ and $\mathbf{F}_E = \mathbf{F}_{E0} e^{i\omega t}$ are the displacement vector and the excitation vector of the entire system. $\mathbf{K}_E^c = (k_0 + i\omega c_0)\mathbf{\Psi}_E$, where $\mathbf{\Psi}_E$ is the coupling matrix (see Source Data file for the details). Equation (9) can thus be written as

$$\left[ \mathbf{K}_E^c - \omega^2 \mathbf{M}_E^{\text{eff}} \right] \mathbf{A}_E = \mathbf{F}_{E0}. \qquad (10)$$

In the following analysis, let $\mathbf{\Phi}_E = \mathbf{K}_E^c - \omega^2 \mathbf{M}_E^{\text{eff}}$. Due to the fixed boundary conditions (some elements in $\mathbf{A}_E$ are zero), the corresponding rows and columns in $\mathbf{\Phi}_E$, $\mathbf{A}_E$, and $\mathbf{F}_{E0}$ must be deleted. The modified matrix and vectors are denoted by $\hat{\mathbf{\Phi}}_E$, $\hat{\mathbf{A}}_E$, and $\hat{\mathbf{F}}_{E0}$. Thus, the displacement amplitude vector of the entire system can be given by

$$\hat{\mathbf{A}}_E = \hat{\mathbf{\Phi}}_E^{-1} \hat{\mathbf{F}}_{E0}. \qquad (11)$$

The transmission of the entire coupling system can be obtained by calculating the ratio of the displacement amplitudes for the output and the input.

The transmissions of the coupling network and the entire coupling system obtained by the above analytical method are shown in Figs. 1b and 1d of the main text, which agrees well with the numerical results.

**Experimental setup**. The experimental setup to investigate the metamaterial system is shown in Fig. 3a. The main body of the metamaterial is fabricated with PLA by using fused-filament-fabrication 3D printing. The hexagon socket-head cap bolts (M4 × 8) and nuts in the metamaterial are made of 304 stainless steel. The metamaterial is fixed on the holders that are glued to the dusk caps of speakers. The speakers can be driven by a power amplifier. An impulse force hammer (Kistler, 9724 A) is used to obtain the transmissions of the metamaterial system. The vibration signals are picked up by an acceleration sensor (B&K, 4371). The measured data is acquired by a data acquisition device (NI, 9234). The transmissions of the metamaterial system are calculated as $|A_{\text{out}}(\omega)|/|F_{\text{in}}(\omega)|$, where $A_{\text{out}}(\omega)$ and $F_{\text{in}}(\omega)$ are the fast Fourier transform amplitudes of the signals from the acceleration sensor and the hammer, respectively. Then, we use a laser vibrometer (Polytec, PSV-500-3D-H) to obtain the vibration modes of the metamaterial.

For the identification tasks, we construct 20 testing signals as a signal set to verify the performance of the metamaterial system. Each signal is constructed by superposing randomized sine waves as $s(t) = \Sigma_{i=1}^{20} a_i \sin(2\pi f_i t + \varphi_i)$, where $t \in [0, 1]$, $a_i \in [0, 1]$, $f_i \in [200, 800]$, and $\varphi_i \in [0, 2\pi]$. Energy normalization is performed by dividing the root mean square of each signal to ensure that the testing signals have the same intensity. The waveforms and spectra of the 20 testing signals are provided in Supplementary Fig. 7. A laptop is used to generate the testing signals. Then, the signals are sent to an integrated acoustic card (M-Audio, M-Track Eight) and amplified by a power amplifier to drive the speakers.

**Construction of the measurement matrix in frequency domain**. The measurement matrix contains the spatial vibration encoding information of the metamaterial and the content of the testing signals. The construction of the measurement matrix includes two steps: experimental calibration and dimensionality compression.

Experimental calibration is currently an effective way to obtain the accurate transmissions due to the unavoidable manufacturing tolerance and the complex testing environments. Theoretically, the measurement matrix $\mathbf{M}_{p \times q}$ can be expressed by $M_q(\omega_p) = H_k(\omega_p) \cdot S_j(\omega_p)$, where $p$ is the index of frequency, $k$ is the index of the location, $j$ is the index of the testing signal, $q = k \times j$ is the column number of measurement matrix, $H_k(\omega_p)$ is the transmission of the metamaterial system, and $S_j(\omega_p)$ is the spectrum of the testing signal. In our experiments, the measurement matrix is calibrated by successively playing the testing signals from different locations and calculating the spectra of the measured signals. More details can be found in Supplementary Note 3.

The current calibrated measurement matrix $\mathbf{M}_{p \times q}$ contains redundant information as well as the noise. After the mixed vibration signal $\mathbf{y}_{p \times 1}$ (i.e., observation vector) is acquired, we respectively replicates the matrices $\mathbf{y}_{p \times 1}$ and $\mathbf{M}_{p \times q}$ by row for $r$ times ($r > 1$ and $r \in \mathbb{Z}$) to increase the weight of the effective information before using dimensionality compression algorithms. The principal component analysis is used to compress the dimensionality of the measurement matrix. First, $\mathbf{M}_{rp \times q}$ and $\mathbf{y}_{rp \times 1}$ are assembled into matrix $\mathbf{A}_{rp \times (q+1)}$. Then, we perform zero-mean normalization on $\mathbf{A}_{rp \times (q+1)}$. Next, singular value decomposition is performed on $\mathbf{A}\mathbf{A}^T$ to obtain the left singular vector matrix $\mathbf{U}_{rp \times rp}$ and the singular value matrix $\mathbf{D}$. By selecting the first $p'$ principal singular values of $\mathbf{D}$ and transforming $\mathbf{U}_{rp \times rp}$ to $\mathbf{U}_{rp \times p'}$, the compressed measurement matrix can be obtained by $\mathbf{M}_{p' \times q} = \mathbf{U}_{rp \times p'}^T \mathbf{M}_{rp \times q}$. In this way, the principal component analysis can compress the measurement matrix to reduce the computational complexity while maintaining the reconstruction accuracy and further improving the robustness against noises.

## Data availability
The source data underlying Fig. 1b–e, 2c,d, 3b,c, 4a–d,f, g, 5a,b, Supplementary Figs. 1, 4a, b, 5a, b, 6b, 7, 8d, e, 9, 10, 11, 12a, b, 13b, 14a–d, and 15c, d are provided as a Source Data file. All other data that support the findings of this study are available from the corresponding author upon reasonable request.

## Code availability
Source code and processed data are available from the corresponding author upon reasonable request.

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

## Acknowledgements
This work was supported by the National Natural Science Foundation of China under Grant No. 11872244, the National Program for Support of Top-Notch Young Professionals, and the State Key Laboratory of Mechanical System and Vibration under Grant No. MSVZD201902. We would like to thank Bo Jing, Dr. Li Yan, and Ying Liu for assistance with the modal experiments. T.J. thanks Dr. Bei Fan, and Rui Yang for their help and useful discussion.

## Author contributions
Q.H. conceived the idea, initiated the metamaterial model, proposed the randomly coupled resonator dynamics, co-wrote the manuscript, and supervised the entire project. T.J. designed the metamaterial model, performed the theoretical study and numerical simulation, carried out the experiments and data analysis, and co-wrote the manuscript. C.L. assisted with the theoretical study and the experiments. Z.K.P. supervised the theoretical study and co-wrote the manuscript. All authors discussed the results, commented on, and revised the manuscript.

## Competing interests
The authors declare no competing interests.
