## [Peer Review File · Nature Communications]

Reviewers' comments:

Reviewer #1 (Remarks to the Author):

The authors present an interesting system for the detection of vibrations with a single detector. The method uses the theory of compressive sensing and a randomized resonant surface, so that the complex frequency response provides different spatial distributions of the field that can be properly applied to recover the field distributions of different sources.

The paper is well presented and results are convincing, however I do not see this work to satisfy the novelty requirements of NatComs, since the concept of compressive sensing and random metamaterials has been already developed for electromagnetic and acoustic waves in different versions,

-Watts, C. M., Shrekenhamer, D., Montoya, J., Lipworth, G., Hunt, J., - Sleasman, T., ... & Padilla, W. J. (2014). Terahertz compressive imaging with metamaterial spatial light modulators. *Nature Photonics*, 8(8), 605.

- Kruizinga, P., van der Meulen, P., Fedjajevs, A., Mastik, F., Springeling, G., de Jong, N., ... & Leus, G. (2017). Compressive 3D ultrasound imaging using a single sensor. *Science advances*, 3(12), e1701423.

- Xie, Y., Tsai, T. H., Konneker, A., Popa, B. I., Brady, D. J., & Cummer, S. A. (2015). Single-sensor multispeaker listening with acoustic metamaterials. *Proceedings of the National Academy of Sciences*, 112(34), 10595-10598.

Therefore, although this paper is not about acoustics but about vibrations, I think that it should be published in a more specialized journal.

Reviewer #2 (Remarks to the Author):

The paper is well written. On the other hand, the following revisions are needed in each section as follows:

1) The introduction is clear and all the objectives well stated. At the same time, some recent technology developments are missing such as:

_ metamaterials [Inverse-designed metastructures that solve equations, *Science* 363 (6433), 1333-1338, 2019]

_ near-zero-index materials [Near-zero-index wires, *Optics express* 25 (20), 23699-23708, 2017]

_ plasmonics [Plasmonic Optical and Chiroptical Response of Self-Assembled Au Nanorod Equilateral Trimers, *ACS nano*, 2019]

_ metasurface [Curvilinear MetaSurfaces for Surface Wave Manipulation, *Scientific reports* 9 (1), 3107, 2019]

_ graphene [Graphene acoustic plasmon resonator for ultrasensitive infrared spectroscopy, *Nature Nanotechnology* 14, 313-319, 2019]

_ nanoparticles [Electromagnetic Nanoparticles for Sensing and Medical Diagnostic Applications, *Materials* 11 (4), 603, 2018]

It would be beneficial for the reader if authors include such technologies in the introduction section to have a complete picture of the state-of-art.

2) The analytical model is interesting, at the same time it is weak and appear insufficient. Explain in details what are the advantages/disadvantages and similarities/differences of the above-mentioned; and how they can be applied to yours.

3) To explore the device behavior, authors can consider the following interesting electromagnetic phenomena:

_ electric/magnetic currents [Metamaterial-based wideband electromagnetic wave absorber, Optics express 24 (6), 5763-5772, 2016]

_ displacements currents [Hybrid bilayer plasmonic metasurface efficiently manipulates visible light, Science Advances, 2(1), 2016]

_ surface waves [Isotropic and anisotropic surface wave cloaking techniques, Journal of Optics 18 (4), 044005]

Include such phenomena in your model and explain how they can affect the device properties.

4) The paper lack in application examples. Take into consideration the following: sensing and diagnostics, biomedicine, telecommunications, absorbers, measurements, nanoelectronics, automotive. I would suggest to create a small paragraph by considering such applications and explaining how you can use your device for them.

Please highlight what's new in yours.

5) No limitations of the proposed method have been highlighted.

6) No future improvements/works have been discussed.

Reviewer #3 (Remarks to the Author):

In the manuscript, the authors present a randomized metamaterial system for single-sensor directional identification of vibration sources. The authors employed the concept of the compressive sensing from electromagnetism and acoustics into vibration problems of finite structures. Although similar results have been demonstrated in acoustics (Ref. 15), the identification of vibration sources is relatively new and interesting. The entire manuscript is well organized and the contents are scientifically sound. While, the reviewer still found some points which need to be further clarified.

1. The authors describe that metamaterials proposed are "anisotropic" in the entire manuscript. However, the effective material properties of the metamaterial should be isotropic. The reviewer does not think disordered structures can be called "anisotropic". In this case, the authors may need a better name.

2. How did the authors calculate the cross-correlation coefficients of the directional frequency responses in Supplementary Fig. S2? How did the authors proof the orthogonality of the randomized metamaterial system? Did the authors use the same method to obtain Fig. 3(b)? What are the definition and physical meaning of the histogram of the cross-correlation coefficients in Fig. 3(c)?

3. The authors mentioned "The correlation of the directional frequency responses (200 ~ 800 Hz) (Fig. 3(b)) ..." and "The signals played each time are randomly selected from the signal set.". The reviewer wonders what is the signal set, and what are the forms of signals used in Figs. 4(a) – 4(d) to examine the devices?

4. The supplementary information only has one paragraph, and the Method section only contains simple explanations on "Construction of the measurement matrix" and "Dimensionality compression of the measurement matrix". As a general reader, the reviewer found difficult to reproduce the results demonstrated in the manuscript. The reviewer feels signal processing part may be more important

than the conceptual part. Because the concept is not totally new. The technique part may be an important contribution, if the authors can provide detailed information for this judgement. Taking an example to demonstrate this in details may be a good strategy.

5. Is specially designed disorder better than the randomized metamaterial? Firstly, the specially designed disorder can maximize the orthogonality with minimum quantities of metamaterial unit cells. Second, the specially designed disorder can be analyzed, which could avoid pre-experimental calibrations.

6. What are the difficulties to increase the source locations from six to larger quantities? Is this physically realizable?

7. In Fig. 2(a), the simplified model contains dampers. What are the damping effects on the identification?

Reviewer's Comment:

The authors present an interesting system for the detection of vibrations with a single detector. The method uses the theory of compressive sensing and a randomized resonant surface, so that the complex frequency response provides different spatial distributions of the field that can be properly applied to recover the field distributions of different sources.

The paper is well presented and results are convincing, however I do not see this work to satisfy the novelty requirements of NatComs, since the concept of compressing sensing and random metamaterials has been already developed for electromagnetic and acoustic waves in different versions,

-Watts, C. M., Shrekenhamer, D., Montoya, J., Lipworth, G., Hunt, J., - Sleasman, T., ... & Padilla, W. J. (2014). Terahertz compressive imaging with metamaterial spatial light modulators. *Nature Photonics*, 8(8), 605.

- Kruizinga, P., van der Meulen, P., Fedjajevs, A., Mastik, F., Springeling, G., de Jong, N., ... & Leus, G. (2017). Compressive 3D ultrasound imaging using a single sensor. *Science advances*, 3(12), e1701423.

- Xie, Y., Tsai, T. H., Konneker, A., Popa, B. I., Brady, D. J., & Cummer, S. A. (2015). Single-sensor multispeaker listening with acoustic metamaterials. *Proceedings of the National Academy of Sciences*, 112(34), 10595-10598.

Therefore, although this paper is not about acoustics but about vibrations, I think that it should be published in a more specialized journal.

Authors' Response:

We thank the reviewer for the careful review and comments on our work. We want to point out that vibrations can easily provide the physical information that electromagnetic and acoustic waves are difficult to reflect to some extent, such as some human vital signs and device conditions. Although the concept of compressing sensing and random metamaterials has been developed for electromagnetic and acoustic waves, there is a knowledge gap in how to design an eligible metamaterial to identify multiple-source vibration information with a single sensor, especially in the low frequency range. To our knowledge, this is the first report showing the spatial vibration encoding and single-sensor identification of elastic vibrations in the metamaterial layer. The proposed randomized resonant metamaterial can reduce the complexity of hardware and the power consumption, which has broad application prospects in health care monitoring, smart devices, and Internet of Things.

We have systematically revised the structure of the manuscript. The revisions are marked by **BLUE** color. First, we revised the abstract and introduction to introduce our concept in a broader background. Then, we developed a new general metamaterial model (**Fig. 1**) to theoretically demonstrate the feasibility of spatial vibration encoding. The general metamaterial model is effective and intuitive, which

provides the physical basis for designing the actual randomized resonant metamaterial (please refer to the section “General metamaterial model for spatial vibration encoding”). Next, to show the practical usefulness of our approach, we added new experiments to demonstrate the novel application of the proposed model (Fig. 5). The results demonstrate that the proposed metamaterial system can be used for tracking trajectory of vibration events, which can be used as a new type of human-machine interface for instruction, communication, and encryption without complex hardware. Moreover, it has potential application prospects in fields such as robot tactile sensing and collision tracking.

Fig. 1. General metamaterial model for highly uncorrelated vibration encoding. (a) Schematic of a simplified supercell network. (b) Transmissions of the OSN and the RSN calculated by analytical and numerical methods. (c) The entire coupled system composed of six different RSNs for vibration identification. The colors of the unit cells denote the resonant frequencies. (d) Transmissions of the coupled system when excited at two terminals respectively. (e) Vibration modes of the coupled system excited at different terminals at 600 Hz.

Fig. 5. Trajectory tracking of vibration events. (a) Experimental setup for trajectory tracking with eight probes. The probes are tapped along the trajectory ‘A-E-F-B-H-D-C-G-A’, and the vibration signals are measured by the single sensor. The trajectory is successfully tracked according to the identification of vibration events. (b) Reconstruction results of vibration events. The dynamic process can be found in Supplementary Video 1. (c) The tracked trajectories of ‘SJTU’. (d) The tracked trajectory ‘Vase’ with 12 probes. The dynamic process can be found in Supplementary Video 2.

Our study can not only open attractive avenues for single-sensor vibration sensing in various fields, but also inspire the design of many simpler sensing systems for other physical information. We believe that our findings can attract broad interest, which would appeal to the readership of *Nature Communications*. We hope that the reviewer can reconsider the novelty and generality of our work, and hope these revisions are satisfactory.

Reviewer's Comment:

The paper is well written. On the other hand, the following revisions are needed in each section as follows:

Authors' Response:

We thank the reviewer for the positive and constructive comments on our work that helped us improve the quality of this manuscript. All the comments and suggestions have been carefully considered and the responses are provided as follows in detail. The revisions in the revised manuscript are marked in BLUE color.

Reviewer's Comment:

1) The introduction is clear and all the objectives well stated. At the same time, some recent technology developments are missing such as:

_ metamaterials [Inverse-designed metastructures that solve equations, *Science* 363 (6433), 1333-1338, 2019]

_ near-zero-index materials [Near-zero-index wires, *Optics express* 25 (20), 23699-23708, 2017]

_ plasmonics [Plasmonic Optical and Chiroptical Response of Self-Assembled Au Nanorod Equilateral Trimers, *ACS nano*, 2019]

_ metasurface [Curvilinear MetaSurfaces for Surface Wave Manipulation, *Scientific reports* 9 (1), 3107, 2019]

_ graphene [Graphene acoustic plasmon resonator for ultrasensitive infrared spectroscopy, *Nature Nanotechnology* 14, 313–319, 2019]

_ nanoparticles [Electromagnetic Nanoparticles for Sensing and Medical Diagnostic Applications, *Materials* 11 (4), 603, 2018]

It would be beneficial for the reader if authors include such technologies in the introduction section to have a complete picture of the state-of-art.

Authors' Response:

We thank the reviewer for the positive and constructive comments. We have included the recent technology developments in the introduction of the revised manuscript as:

“Metamaterials are a broad family of artificially structured materials with unusual effective properties and functionalities²¹⁻²⁵. Flexible manipulations of electromagnetic, acoustic, and elastic waves can be achieved, such as cloaking²⁶⁻²⁸, beaming^{29,30}, diffusing³¹, illusion³², and hologram³³. Fascinating applications, such as high-speed analog computing^{34,35}, ultrasensitive detection³⁶⁻³⁸, and efficient waveguiding³⁹ have been demonstrated.

21. Liu, Z. *et al.* Locally resonant sonic materials. *Science* **289**, 1734-1736 (2000).

22. Ma, G. & Sheng, P. Acoustic metamaterials: From local resonances to broad horizons. *Sci. Adv.* **2**, e1501595 (2016).

23. Ge, H. *et al.* Breaking the barriers: advances in acoustic functional materials. *National Science Review* **5**, 159-182 (2017).
24. Assouar, B. *et al.* Acoustic metasurfaces. *Nat. Rev. Mater.*, 1 (2018).
25. Greybush, N. J., Pacheco-Peña, V., Engheta, N., Murray, C. B. & Kagan, C. R. Plasmonic Optical and Chiroptical Response of Self-Assembled Au Nanorod Equilateral Trimers. *ACS Nano* **13**, 1617-1624 (2019).
26. Zhang, S., Xia, C. & Fang, N. Broadband acoustic cloak for ultrasound waves. *Phys. Rev. Lett.* **106**, 024301 (2011).
27. Zigoneanu, L., Popa, B.-I. & Cummer, S. A. Three-dimensional broadband omnidirectional acoustic ground cloak. *Nat. Mater.* **13**, 352 (2014).
28. La Spada, L., Spooner, C., Haq, S. & Hao, Y. Curvilinear metasurfaces for surface wave manipulation. *Sci. Rep.* **9**, 3107 (2019).
29. Shen, C., Xu, J., Fang, N. X. & Jing, Y. Anisotropic complementary acoustic metamaterial for canceling out aberrating layers. *Phys. Rev. X* **4**, 041033 (2014).
30. Foehr, A., Bilal, O. R., Huber, S. D. & Daraio, C. Spiral-based phononic plates: From wave beaming to topological insulators. *Phys. Rev. Lett.* **120**, 205501 (2018).
31. Zhu, Y., Fan, X., Liang, B., Cheng, J. & Jing, Y. Ultrathin acoustic metasurface-based Schroeder diffuser. *Phys. Rev. X* **7**, 021034 (2017).
32. Liu, Y. *et al.* Source illusion devices for flexural Lamb waves using elastic metasurfaces. *Phys. Rev. Lett.* **119**, 034301 (2017).
33. Zhu, Y. *et al.* Fine manipulation of sound via lossy metamaterials with independent and arbitrary reflection amplitude and phase. *Nat. Commun.* **9**, 1632 (2018).
34. Estakhri, N. M., Edwards, B. & Engheta, N. Inverse-designed metastructures that solve equations. *Science* **363**, 1333-1338 (2019).
35. Zangeneh-Nejad, F. & Fleury, R. Topological analog signal processing. *Nat. Commun.* **10**, 2058 (2019).
36. Lee, I.-H., Yoo, D., Avouris, P., Low, T. & Oh, S.-H. Graphene acoustic plasmon resonator for ultrasensitive infrared spectroscopy. *Nat. Nanotechnol.* **14**, 313 (2019).
37. La Spada, L. & Vegni, L. Electromagnetic nanoparticles for sensing and medical diagnostic applications. *Materials* **11**, 603 (2018).
38. Zhang, Z. *et al.* Directional Acoustic Antennas Based on Valley-Hall Topological Insulators. *Adv. Mater.* **30**, 1803229 (2018).
39. La Spada, L. & Vegni, L. Near-zero-index wires. *Opt. Express* **25**, 23699-23708 (2017).”

(Page 1, Left column, Line 48)

Reviewer's Comment:

2) The analytical model is interesting, at the same time it is weak and appear insufficient. Explain in details what are the advantages/disadvantages and similarities/differences of the above-mentioned; and how they can be applied to yours.

Authors' Response:

Thanks for the reviewer's constructive comments. The above-mentioned references are listed at the end of this response. Ref. [1] mainly uses the lumped circuit model to design the metasurface, which is partly similar to our proposed elastic metamaterial. Ref. [2] also uses the effective medium theory to describe the properties of the electromagnetic nanoparticles. Although electromagnetic metamaterials are not our research fields, we have strengthened the analytical model inspired by the effective models in Refs. [1] and [2] provided by the reviewer. We have developed a general elastic metamaterial model by simplifying the locally resonant unit cell as an effective mass. The revised

analytical model is more accurate than the original one. The proposed general elastic metamaterial model is effective to demonstrate the highly uncorrelated vibration transmissions and the complexity of the elastic vibration propagation, which provides the physical basis for designing the actual randomized resonant metamaterial system and is easy to be extended. We moved the section “General metamaterial model for spatial vibration encoding” to the first section of Results to better organize our manuscript. The related revisions are presented as follows:

“A general metamaterial model is proposed to achieve the highly uncorrelated transmissions for spatial vibration information encoding. The proposed model consists of multiple different supercell networks as shown in Fig. 1(a). Each supercell network is composed of N locally resonant unit cells connected by springs k_0 and dampers c_0 . k_n and c_n are the stiffness and the damping coefficients of the n th unit cell, respectively. x_n and y_n are displacements of matrix M and mass m . The locally resonant unit cell can be simplified as an effective mass m_n^{eff} , which is expressed as

$$m_n^{\text{eff}} = M + \frac{k_n^{\text{d}} m}{k_n^{\text{d}} - \omega^2 m}, \quad (1)$$

where $k_n^{\text{d}} = k_n + i\omega c_n$, $k_n = m\omega_n^2$, and ω_n is the resonant frequency of the n th unit cell. The effective mass can be negative near the resonant frequency due to the anti-resonance effect, which corresponds to the attenuation of elastic waves (see Methods and Supplementary Note 1).

In the supercell network, the effective masses of all unit cells can be the same or different. For convenience, we refer to the supercell network with the same unit cells as an ordinary supercell network (OSN) and that with random unit cells as a randomized supercell network (RSN). By using the finite element method (FEM) and the analytical method, the transmissions of an OSN (with a local resonance frequency of 420 Hz) and a RSN can be obtained (Fig. 1(b)) (see Methods for details of the numerical simulation and the analytical derivation). The analytical results agree well with the numerical results. For the OSN, the transmission has a wide attenuation region (gray shaded region in Fig. 1(b)) due to the presence of the local resonance band gap. If the OSN is used for sensing, the information of the vibrations in the band gap will be lost, which is disadvantageous for vibration encoding and identification. In contrast, the RSN can randomly encode vibration information in the frequency domain because the randomized effective masses of the unit cells lead to the random distributions of the amplification and attenuation bands. This randomized encoding property is useful for constructing a desirable measurement matrix for the compressive sensing.

Figure 1(c) shows an entire coupled system by connecting six RSNs and a center module (without locally resonant unit cells) for spatial vibration encoding. In each RSN, the spatial distribution is disordered for the effective masses of unit cells. Vibrations generated from multiple sources propagate from the boundaries to the center module. The transmission of the coupled system can be obtained by measuring the vibration signals of the center mass M_0 . Figure 1(d) shows the transmissions of the coupled system when excitations are applied at two different locations (see Supplementary Fig. 4(a) for transmissions corresponding to the other locations). It can be seen that substantial differences exist between the two transmissions. ... We choose the average of the absolute cross-correlation coefficients μ_{Ave} as the metric of the uncorrelation. Here, the μ_{Ave} is approximately equal to 0.15, indicating that the transmissions are desirably uncorrelated.

To explain the relationship between the uncorrelation and the disordered effective masses, we analyze the vibration modes of the entire coupled system. Figure 1(e) shows the vibration modes obtained by respectively applying excitations to six locations at 600 Hz. Due to the disordered distribution of the random effective masses, the displacement fields are obviously different, which reflects

the complexity of the elastic vibration propagation in the coupled system. The disordered effective masses leads to the high uncorrelation of the transmissions, which is the physical basis for effective spatial vibration encoding.” (Page 1, Right column, Line 81)

Fig. 1. General metamaterial model for spatial vibration encoding. (a) Schematic of a simplified supercell network. (b) Transmissions of the OSN and the RSN calculated by analytical and numerical methods. (c) The entire coupled system composed of six different RSNs for vibration identification. The colors of the unit cells denote the resonant frequencies. (d) Transmissions of the coupled system when excited at two terminals respectively. (e) Vibration modes of the coupled system excited at different terminals at 600 Hz.

The analytical derivation of the revised general metamaterial model are presented as follows:

“The dynamical equation of the n th unit cell can be expressed as

$$\begin{cases} M\ddot{x}_n + c_0 \left(\sum_{i=1}^N \psi_{ni} \dot{x}_i \right) + k_0 \left(\sum_{i=1}^N \psi_{ni} x_i \right) - c_n (\dot{y}_n - \dot{x}_n) - k_n (y_n - x_n) = F_n, \\ m\ddot{y}_n + c_n (\dot{y}_n - \dot{x}_n) + k_n (y_n - x_n) = 0, \end{cases} \quad (2)$$

where

$$\psi_{ni} = \begin{cases} -1, & i \text{ couples with } n, \\ \phi_n, & i = n, \\ 0, & \text{others,} \end{cases} \quad (3)$$

ϕ_n is equal to the number of the unit cells coupling with the n th unit cell. Eq. 2 can be derived as

$$\left(M + \frac{k_n^d m}{k_n^d - \omega^2 m} \right) \ddot{x}_n + (k_0 + i\omega c_0) \left(\sum_{i=1}^N \psi_{ni} x_i \right) = F_n \quad (4)$$

The expression of the effective mass (Eq. (1)) can thus be obtained. From Eq. (4), we can express the dynamical equation of the k th supercell network as a matrix-vector form of

$$\mathbf{M}_k^{\text{eff}} \ddot{\mathbf{X}}_k + \mathbf{K}_k^c \mathbf{X}_k = \mathbf{F}_k, \quad (5)$$

where $\mathbf{M}_k^{\text{eff}} = \text{diag}(m_1^{\text{eff}}, m_2^{\text{eff}}, \dots, m_N^{\text{eff}})$ is a diagonal matrix composed of effective masses given by Eq.

(1), and N is the number of the unit cells. The displacement vector of the unit cells has the form of $\mathbf{X}_k = [x_1, x_2, \dots, x_N]^T = \mathbf{A}_k e^{i\omega t}$, where \mathbf{A}_k is the amplitude vector. The stiffness-damping matrix \mathbf{K}_k^c can be expressed as

$$\mathbf{K}_k^c = (k_0 + i\omega c_0)\Psi_k, \quad (6)$$

where Ψ_k with the elements ψ_{ni} is a $N \times N$ coupling matrix reflecting the coupling relationship of the unit cells (see Source Data file for the details). The time-harmonic excitation applied to the unit cells is $\mathbf{F}_k = [F_1, F_2, \dots, F_N]^T = \mathbf{F}_{k0} e^{i\omega t}$. Thus, the displacement amplitude vector \mathbf{A}_k can be calculated by

$$\mathbf{A}_k = [\mathbf{K}_k^c - \omega^2 \mathbf{M}_k^{\text{eff}}]^{-1} \mathbf{F}_{k0}. \quad (7)$$

Therefore, the transmission of the supercell network can be expressed as

$$T_{\text{Amp}} = |x_{\text{out}}|/|x_{\text{in}}|, \quad (8)$$

where $|x_{\text{out}}|$ and $|x_{\text{in}}|$ are the elements in \mathbf{A}_k that correspond to the output and the input of the supercell network.

The transmission of the entire coupled system can also be derived in a similar way. The dynamical equation of the coupled system is given by

$$\mathbf{M}_E^{\text{eff}} \ddot{\mathbf{X}}_E + \mathbf{K}_E^c \mathbf{X}_E = \mathbf{F}_E. \quad (9)$$

Here, $\mathbf{M}_E^{\text{eff}} = \text{diag}(\mathbf{M}_1^{\text{eff}}, \mathbf{M}_2^{\text{eff}}, \dots, \mathbf{M}_6^{\text{eff}}, \mathbf{M}_0)$, where $\mathbf{M}_0 = \text{diag}(M, \dots, M)$ is the mass matrix of the center module of the system. $\mathbf{X}_E = \mathbf{A}_E e^{i\omega t}$ and $\mathbf{F}_E = \mathbf{F}_{E0} e^{i\omega t}$ are the displacement vector and the excitation vector of the entire system. $\mathbf{K}_E^c = (k_0 + i\omega c_0)\Psi_E$, where Ψ_E is the coupling matrix (see Source Data file for the details). Eq. (9) can thus be written as

$$[\mathbf{K}_E^c - \omega^2 \mathbf{M}_E^{\text{eff}}] \mathbf{A}_E = \mathbf{F}_{E0}. \quad (10)$$

In the following analysis, let $\Phi_E = \mathbf{K}_E^c - \omega^2 \mathbf{M}_E^{\text{eff}}$. Due to the fixed boundary conditions (some elements in \mathbf{A}_E are zero), the corresponding rows and columns in Φ_E , \mathbf{A}_E , and \mathbf{F}_{E0} must be deleted. The modified matrix and vectors are denoted by $\hat{\Phi}_E$, $\hat{\mathbf{A}}_E$, and $\hat{\mathbf{F}}_{E0}$. Thus, the displacement amplitude vector of the entire system can be given by

$$\hat{\mathbf{A}}_E = \hat{\Phi}_E^{-1} \hat{\mathbf{F}}_{E0}. \quad (11)$$

The transmission of the entire coupled system can be obtained by calculating the ratio of the displacement amplitudes for the output and the input.

The transmissions of the supercell network and the entire coupled system obtained by the above analytical method is shown in Figs. 1(b) and 1(d) of the main text, which agrees well with the numerical results.” (Methods: Analytical derivation of the general metamaterial model)

“Supplementary Fig. 1 shows the effective mass of a unit cell with the resonant frequency of 541 Hz calculated by Eq. (1). It can be seen that the real part and imaginary part of the effective mass are negative near the resonant frequency, which means that the vibration decays exponentially as it propagates.” (Supplementary Note 1)

Supplementary Figure 1. Effective mass of a unit cell with the resonant frequency of 541 Hz.

Ref. [3] is mainly based on the scattering mechanism rather than the resonance mechanism, which is different from our proposed model for the low frequency case. Nonetheless, we believe that the proposed methods in Refs. [1] and [3] are beneficial to the optimization design and the performance prediction of the device. We have added the discussion in the revised manuscript, and the related statements are presented as follows:

“Although the proposed metamaterial system is a prototype, it provides an attractive avenue for simpler vibration sensing, and it can be further improved in the following aspects. ... In addition, the field theory can be introduced to improve the analytical model for further optimization design and performance prediction of the device^{28,45}.

However, there are also limitations for the proposed approach. First, although the proposed metamaterial system has achieved an effective spatial vibration encoding, the randomized design strategy is not the optimal one that can maximize the uncorrelation while minimizing the number of unit cells. This limitation can be overcome by using topological optimization^{34,46} or machine learning methods⁴⁷ to achieve the on-demand design of the disordered structures in the further work.”

28. La Spada, L., Spooner, C., Haq, S. & Hao, Y. Curvilinear metasurfaces for surface wave manipulation. *Sci. Rep.* **9**, 3107 (2019).

34. Estakhri, N. M., Edwards, B. & Engheta, N. Inverse-designed metastructures that solve equations. *Science* **363**, 1333-1338 (2019).

45. McManus, T., La Spada, L. & Hao, Y. Isotropic and anisotropic surface wave cloaking techniques. *J. Optics* **18**, 044005 (2016).

46. Dong, H.-W., Zhao, S.-D., Wang, Y.-S. & Zhang, C. Topology optimization of anisotropic broadband double-negative elastic metamaterials. *J. Mech. Phys. Solids* **105**, 54-80 (2017).

47. Ma, W., Cheng, F. & Liu, Y. Deep-learning-enabled on-demand design of chiral metamaterials. *ACS Nano* **12**, 6326-6334 (2018).” (Page 6, Left column, Line 8)

We tried our best but were not able to comment on the methods in Refs. [4] ~ [6]. This is because the dielectric wire [4], the graphene acoustic plasmon [5], and the plasmonic optics [6] are not our research fields, and we are not familiar with the theory in the fields. We envisioned that our proposed method can inspire the related study of elastic wave sensing in these fields.

[1] Curvilinear MetaSurfaces for Surface Wave Manipulation, Scientific reports 9 (1), 3107, 2019

[2] Electromagnetic Nanoparticles for Sensing and Medical Diagnostic Applications, Materials 11 (4), 603, 2018

- [3] Inverse-designed metastructures that solve equations, *Science* 363 (6433), 1333-1338, 2019
- [4] Near-zero-index wires, *Optics express* 25 (20), 23699-23708, 2017
- [5] Plasmonic Optical and Chiroptical Response of Self-Assembled Au Nanorod Equilateral Trimers, *ACS nano*, 2019
- [6] Graphene acoustic plasmon resonator for ultrasensitive infrared spectroscopy, *Nature Nanotechnology* 14, 313–319, 2019

Reviewer's Comment:

3) To explore the device behavior, authors can consider the following interesting electromagnetic phenomena:

_ electric/magnetic currents [Metamaterial-based wideband electromagnetic wave absorber, *Optics express* 24 (6), 5763-5772, 2016]

_ displacements currents [Hybrid bilayer plasmonic metasurface efficiently manipulates visible light, *Science Advances*, 2(1), 2016]

_ surface waves [Isotropic and anisotropic surface wave cloaking techniques, *Journal of Optics* 18 (4), 044005]

Include such phenomena in your model and explain how they can affect the device properties.

Authors' Response:

Thanks for the reviewer's comments. It should be pointed out that our study mainly focuses on the uncorrelated transmission property of elastic vibrations rather than the electromagnetic waves. To explain the relationship between the uncorrelated transmission property and the disordered effective masses, we analyze the vibration modes of the general metamaterial model. Due to the disordered distribution of the random effective masses, the displacement fields are obviously different as shown in revised Fig. 1(e), which reflects the complexity of the elastic vibration propagation in the coupled system. This property leads to the high uncorrelation of the transmissions, which is the physical basis for effective spatial vibration encoding. We also experimentally measured the complex vibration velocity field distributions of the actual metamaterial system as shown in the revised Fig. 3(d). The results visually show the high complexity of the elastic vibration propagation in the metamaterial, which confirms the effective spatial vibration encoding ability of the designed metamaterial. The above modal analysis reflects the elastic vibration propagation property of the proposed metamaterial, which may be similar to the analysis of the electric/magnetic currents [1], the displacement currents [2], the surface waves [3] mentioned by the reviewer. The related discussions have been included in the revised manuscript, and are presented as follows:

“To explain the relationship between the uncorrelation and the disordered effective masses, we analyze the vibration modes of the entire coupled system. Figure 1(e) shows the vibration modes obtained by respectively applying excitations to six locations at 600 Hz. Due to the disordered distribution of the random effective masses, the displacement fields are obviously different, which reflects the complexity of the elastic vibration propagation in the coupled system. The disordered effective masses leads to the high uncorrelation of the transmissions, which is the physical basis for effective spatial vibration encoding.” (Page 2, Right column, Line 36)

“Figure 3(d) specifically presents the complex vibration velocity field distributions at 695.3 Hz, which are experimentally measured by applying excitations to different locations. The results visually

show the high complexity of the elastic vibration propagation in the metamaterial, which confirms the effective spatial vibration encoding ability of the designed metamaterial.” (Page 3, Right column, Line 59)

Fig. 1. General metamaterial model for spatial vibration encoding. (a) Schematic of a simplified supercell network. (b) Transmissions of the OSN and the RSN calculated by analytical and numerical methods. (c) The entire coupled system composed of six different RSNs for vibration identification. The colors of the unit cells denote the resonant frequencies. (d) Transmissions of the coupled system when excited at two terminals respectively. (e) Vibration modes of the coupled system excited at different terminals at 600 Hz.

Fig. 3. Spatial vibration encoding property of the metamaterial system. (a) Experimental setup to test the metamaterial system. (b) Transmissions when vibration sources are excited at six different locations. (c) Correlation of the transmissions. (d) Vibration modes of the metamaterial when excited at different locations at 695.3 Hz.

[1] Metamaterial-based wideband electromagnetic wave absorber, Optics express 24 (6), 5763-5772, 2016

[2] Hybrid bilayer plasmonic metasurface efficiently manipulates visible light, Science Advances, 2(1), 2016

Reviewer's Comment:

4) The paper lack in application examples. Take into consideration the following: sensing and diagnostics, biomedicine, telecommunications, absorbers, measurements, nanoelectronics, automotive. I would suggest to create a small paragraph by considering such applications and explaining how you can use your device for them.

Please highlight what's new in yours.

Authors' Response:

Thanks for the reviewer's constructive comments and suggestions. To illustrate the usefulness of our approach, we demonstrated that the proposed device can achieve the trajectory tracking of vibration events by adding new experiments. The results are provided in the revised Fig. 5, and Supplementary Videos 1 and 2. The results indicate that the proposed device can be used as a new type of human-machine interface for instruction, communication, and encryption without complex hardware and high power consumption. We also envisioned that this device has potential application prospects in fields such as robot tactile sensing and collision tracking. Moreover, we discussed the improvements to further improve the usefulness of our device: "The size of the system can be further reduced with the help of advanced manufacturing techniques and microelectronics, so that it can be flexibly integrated into numerous smart devices." (Page 6, Left column, Line 15) The new paragraph for demonstrating the applications has been added in the revised manuscript, and specifically presented as follows:

"To exhibit the broad application prospects, we demonstrate a use of the proposed metamaterial system for trajectory tracking of vibration events. Here, eight probes (denoted by 'A' to 'H') are fixed around the metamaterial as shown in Fig. 5(a). We tap the probes with the finger in sequence along the illustrated trajectory 'A-E-F-B-H-D-C-G-A', and the vibration signals are measured as shown in Fig. 5(a) by the single sensor. We can clearly see the tracked trajectory according to the identification of vibration events. Figure 5(b) shows the reconstructed results of each vibration event as well as the trajectory pattern, where the occurrence locations and time of events agree well with the truth (see Supplementary Note 10 for details). This trajectory tracking process is dynamically presented in Supplementary Video 1. We also study the tracking process of the trajectories 'SJTU' (i.e. the abbreviation of Shanghai Jiao Tong University). The tracked trajectories are well exhibited in Fig. 5(c) (see Supplementary Fig. 14 for the details). Figure 5(d) shows the successfully tracked trajectory 'Vase' with 12 probes. The experimental details and reconstruction results are presented in Supplementary Fig. 15 and Supplementary Video 2. The results above demonstrate that the proposed metamaterial system can be used as a new type of human-machine interface for instruction, communication, and encryption without complex hardware and high power consumption. Moreover, it also has potential application prospects in fields such as robot tactile sensing and collision tracking." (Page 5, Left column, Line 23)

Fig. 5. Trajectory tracking of vibration events. (a) Experimental setup for trajectory tracking with eight probes. The probes are tapped along the trajectory ‘A-E-F-B-H-D-C-G-A’, and the vibration signals are measured by the single sensor. The trajectory is successfully tracked according to the identification of vibration events. (b) Reconstruction results of vibration events. The dynamic process can be found in Supplementary Video 1. (c) The tracked trajectories of ‘SJTU’. (d) The tracked trajectory ‘Vase’ with 12 probes. The dynamic process can be found in Supplementary Video 2.

“The waveform presented in Fig. 5(a) is the whole measured signal in the trajectory tracking. We continuously intercept the measured signal according to each vibration event as shown in Supplementary Fig. 13(a). Here we take an example to show the identification performance, where the 7th fragment of the measured signal is used to be the observation vector. Supplementary Fig. 13(b) shows the normalized reconstruction result of the selected vibration event. The occurrence location and time of the reconstructed impact are in good agreement with the actual one. We calculate the maximum value of the reconstructed vector segment corresponding to each location in Supplementary Fig. 13(b). The maximum values are visualized in Supplementary Fig. 13(c). It can be seen that an impact is applied to the probe C. Furthermore, we use a similar method to reconstruct the trajectories ‘SJTU’ as shown in Supplementary Fig. 14.

Supplementary Fig. 15 shows the reconstruction details of the trajectory ‘Vase’, where the probes are increased to 12. It can be seen that the occurrence locations and time of the vibration events can still be successfully identified. This tracking process is dynamically shown in Supplementary Video 2. The results above demonstrate that the proposed device can be used to track complex trajectories, which has potential application prospects in fields such as human-machine interface and collision tracking.” (Supplementary Note 10)

Supplementary Figure 13. Details of the reconstruction for trajectory tracking. (a) The measured signal from the single sensor. (b) The normalized reconstruction results of the 7th vibration event, and (c) the visualization of the maximum values in the reconstructed vector segments.

Supplementary Figure 14. Tracking process of the trajectories 'SJTU'. (a)-(d) The tracked trajectories, measured signals, and reconstruction results of 'S', 'J', 'T', and 'U', respectively.

Supplementary Figure 15. Tracking process of the trajectory ‘Vase’. (a) Experimental setup. (b) The tracked trajectory. (c) The measured signal from the single sensor. (d) Reconstruction results of each vibration event.

Reviewer’s Comment:

5) No limitations of the proposed method have been highlighted.

Authors’ Response:

Thanks for the reviewer’s constructive comment. We have highlighted the limitations of the proposed method in the discussion, and pointed out the possible solutions to overcome the limitations in the future work. The related discussions are specifically presented as follows:

“However, there are also limitations for the proposed approach. First, although the proposed metamaterial system has achieved an effective spatial vibration encoding, the randomized design strategy is not the optimal one that can maximize the uncorrelation while minimizing the number of unit cells. This limitation can be overcome by using topological optimization^{34,46} or machine learning methods⁴⁷ to achieve the on-demand design of the disordered structures in the further work. Moreover, the proposed approach needs the pre-experimental calibrations because the actual sample in experiments might not match the designed one due to the unavoidable manufacturing tolerance. This issue is hoped to be solved by utilizing artificial intelligence algorithms based on the computational and experimental data^{48,49}.

34. Estakhri, N. M., Edwards, B. & Engheta, N. Inverse-designed metastructures that solve equations. *Science* **363**, 1333-1338 (2019).

46. Dong, H.-W., Zhao, S.-D., Wang, Y.-S. & Zhang, C. Topology optimization of anisotropic broadband double-negative elastic metamaterials. *J. Mech. Phys. Solids* **105**, 54-80 (2017).

47. Ma, W., Cheng, F. & Liu, Y. Deep-learning-enabled on-demand design of chiral metamaterials. *ACS Nano* **12**, 6326-6334 (2018).

48. Wang, F., Wang, H., Wang, H., Li, G. & Situ, G. Learning from simulation: An end-to-end deep-learning approach for computational ghost imaging. *Opt. Express* **27**, 25560-25572 (2019).

49. Jha, D. *et al.* Enhancing materials property prediction by leveraging computational and experimental data using

deep transfer learning. *Nat. Commun.* **10**, 1-12 (2019).” (Page 6, Left column, Line 19)

Reviewer’s Comment:

6) No future improvements/works have been discussed.

Authors’ Response:

Thanks for the reviewer’s constructive comment. In the Author’s Response for Reviewer’s Comment 5, we have discussed the future improvements and works which can overcome the limitations of the proposed method, including optimization design and introducing artificial intelligence algorithms. In addition, we have added more discussion in the revised manuscript to further improve the proposed metamaterial system. The discussions are specifically presented as follows:

“Although the proposed metamaterial system is a prototype, it provides an attractive avenue for simpler vibration sensing, and it can be further improved in the following aspects. The metamaterial can be assembled into more configurations to identify potential spatial vibrations according to practical needs, e.g., a 3D structure configuration⁴⁴ to identify omnidirectional elastic vibrations. The operational frequency ranges can be adjusted by changing the structural parameters of unit cells. The size of the system can be further reduced with the help of advanced manufacturing techniques and microelectronics, so that it can be flexibly integrated into numerous smart devices. In addition, the field theory can be introduced to improve the analytical model for further optimization design and performance prediction of the device^{28,45}.

28. La Spada, L., Spooner, C., Haq, S. & Hao, Y. Curvilinear metasurfaces for surface wave manipulation. *Sci. Rep.* **9**, 3107 (2019).

44. Yan, Z. *et al.* Controlled mechanical buckling for origami-inspired construction of 3D microstructures in advanced materials. *Advanced functional materials* **26**, 2629-2639 (2016).

45. McManus, T., La Spada, L. & Hao, Y. Isotropic and anisotropic surface wave cloaking techniques. *J. Optics* **18**, 044005 (2016).” (Page 6, Left column, Line 8)

We would like to thank the reviewer for the positive and constructive review on our work again and hope these revisions are satisfactory.

Reviewer's Comment:

In the manuscript, the authors present a randomized metamaterial system for single-sensor directional identification of vibration sources. The authors employed the concept of the compressive sensing from electromagnetism and acoustics into vibration problems of finite structures. Although similar results have been demonstrated in acoustics (Ref. 15), the identification of vibration sources is relatively new and interesting. The entire manuscript is well organized and the contents are scientifically sound. While, the reviewer still found some points which need to be further clarified.

Authors' Response:

We thank the reviewer very much for the positive and constructive comments on our work that helped us improve the quality of this manuscript. All the comments and suggestions have been carefully considered and the responses are provided as follows in detail. The related revisions of the manuscript are marked in BLUE color.

Reviewer's Comment:

1. The authors describe that metamaterials proposed are “anisotropic” in the entire manuscript. However, the effective material properties of the metamaterial should be isotropic. The reviewer does not think disordered structures can be called “anisotropic”. In this case, the authors may need a better name.

Authors' Response:

We thank the reviewer for pointing out this problem and the constructive comment. We now use the name “randomized resonant metamaterials” to describe the proposed metamaterial, because the resonant frequencies of the unit cells are randomized. In addition, we use “disordered” to describe the spatial arrangement of the effective masses of the randomized unit cells. The title has been revised as “Randomized resonant metamaterials for single-sensor identification of elastic vibrations”, and other related descriptions have also been modified.

Reviewer's Comment:

2. How did the authors calculate the cross-correlation coefficients of the directional frequency responses in Supplementary Fig. S2?

Authors' Response:

Thanks for the reviewer's constructive comment. We have added the calculation of the cross-correlation coefficients in the main text and the revised Supplementary Note 3. The part “Evaluation of the uncorrelation of vibration transmissions” has been revised and specifically presented as follows:

“To evaluate the uncorrelation of the transmissions, we calculate the correlation coefficients and the correlation matrix C_μ (see Supplementary Note 3). The absolute values of elements in C_μ are visualized as shown in Supplementary Fig. 4(b). The histogram of the absolute cross-correlation coefficients show that they narrowly distribute around zero (Supplementary Fig. 4(c)). We choose the

average of the absolute cross-correlation coefficients μ_{Ave} as the metric of the uncorrelation. Here, the μ_{Ave} is approximately equal to 0.15, indicating that the transmissions are desirably uncorrelated.” (Page 2, Left column, Line 25)

“The vibration transmissions of the metamaterial are contributed to the construction of the measurement matrix, which directly influences the performance of the vibration identification. Supplementary Fig. 4(a) shows the transmissions of the coupled system. The correlation coefficients μ_{ij} of the transmissions is calculated as

$$\mu_{ij} = \frac{\sum_p [H_i(\omega_p) - \bar{H}_i][H_j(\omega_p) - \bar{H}_j]}{\sqrt{\sum_p (H_i(\omega_p) - \bar{H}_i)^2} \sqrt{\sum_p (H_j(\omega_p) - \bar{H}_j)^2}}, \quad (S1)$$

where p is the index of the frequency, $H_i(\omega_p)$ and $H_j(\omega_p)$ are transmissions of the coupled system, \bar{H}_i and \bar{H}_j are the averages of $H_i(\omega_p)$ and $H_j(\omega_p)$. Here, we define the correlation matrix \mathbf{C}_μ as

$$\mathbf{C}_\mu = \begin{bmatrix} \mu_{11} & \cdots & \mu_{1k} \\ \vdots & \ddots & \vdots \\ \mu_{k1} & \cdots & \mu_{kk} \end{bmatrix}, \quad (S2)$$

where k is the number of source locations. Supplementary Fig. 4(b) visualizes the absolute values of the elements in \mathbf{C}_μ (i.e. $|\mu_{ij}|$) for the coupled system. To quantify the uncorrelation of the vibration transmissions, the average of the absolute cross-correlation coefficients μ_{Ave} is defined as

$$\mu_{Ave} = \frac{1}{k(k-1)} \sum_{i \neq j} |\mu_{ij}|. \quad (S3)$$

Here, μ_{Ave} ranges from an ideal 0 (i.e. perfectly orthogonal modulation) to a useless 1 (i.e. identical modulation). To avoid the existence of extreme large values (while μ_{Ave} is still small), we plot the histogram to show the distribution of the absolute cross-correlation coefficients in Supplementary Fig. 4(c). In this way, we can make sure that the correlations of each two transmissions are small. In the histogram, the absolute cross-correlation coefficients distribute narrowly around zero, indicating that the transmissions are highly uncorrelated.” (Supplementary Note 3)

Supplementary Figure 4. Transmission property of the entire coupled system. (a) Transmissions of the entire coupled system for six different locations. (b) Absolute correlation coefficients of the transmissions. (c) Histogram of the absolute cross-correlation coefficients.

Reviewer's Comment:

How did the authors proof the orthogonality of the randomized metamaterial system?

Authors' Response:

Thanks for the reviewer's rigorous comment. We used the term "orthogonality" to describe the high uncorrelation of the vibration transmission property in the original manuscript. We have realized that the original expression is not rigorous, although the average of the absolute cross-correlation coefficients μ_{Ave} is small. In the revised manuscript, we have replaced "orthogonality" and the relevant statements with "uncorrelation" or "highly uncorrelated". The revisions are marked in BLUE color.

Reviewer's Comment:

Did the authors use the same method to obtain Fig. 3(b)?

Authors' Response:

Yes. We modified the original Fig. 3(b) as the revised Fig. 3(c). The related statements have been revised as: "Spatial transmissions can be obtained by using the experimental modal analysis method (Fig. 3(b)). The correlation of the transmissions (Fig. 3(c)) is calculated by using the same method in Supplementary Note 3, and the average cross-correlation coefficient μ_{Ave} is achieved as 0.15."

(Page 3, Right column, Line 51)

Fig. 3. Spatial vibration encoding property of the metamaterial system. (a) Experimental setup to test the metamaterial system. (b) Transmissions when vibration sources are excited at six different locations. (c) Correlation of the transmissions. (d) Vibration modes of the metamaterial when excited at different locations at 695.3 Hz.

Reviewer's Comment:

What are the definition and physical meaning of the histogram of the cross-correlation coefficients in Fig. 3(c)?

Authors' Response:

In the revised manuscript, the definition of the histogram of the cross-correlation coefficients is the histogram of the absolute values of the upper triangular elements in C_μ (please refer to the above responses and the revised Supplementary Note 3). The histogram can show the distribution of the correlation, and help us to avoid the values close to 1. In this way, we can make sure that the correlations of each two transmissions are small. When the absolute cross-correlation coefficients are more narrowly distributed around zero, the vibration transmission property becomes more uncorrelated. We added the discussions in the supplementary material, and the related statements are presented as follows:

“To avoid the existence of extreme values close to 1 (while μ_{Ave} is still small), we plot the histogram to show the distribution of the absolute cross-correlation coefficients in Supplementary Fig. 4(c). In this way, we can make sure that the correlations of each two transmissions are small. In the histogram, the absolute cross-correlation coefficients distribute narrowly around zero, indicating that the transmissions are highly uncorrelated.” (Supplementary Note 3)

Reviewer's Comment:

3. The authors mentioned “The correlation of the directional frequency responses (200 ~ 800 Hz) (Fig. 3(b)) ...” and “The signals played each time are randomly selected from the signal set.”. The reviewer wonders what is the signal set, and what are the forms of signals used in Figs. 4(a) – 4(d) to examine the devices?

Authors' Response:

Thanks for the reviewer's constructive comments. We have provided the spatial vibration transmissions (directional frequency responses) in the revised Fig. 3(b). The related statement has been revised as: “Spatial transmissions can be obtained by using the experimental modal analysis method (Fig. 3(b)).” (Page 3, Right column, Line 51)

Fig. 3. Spatial vibration encoding property of the metamaterial system. (a) Experimental setup to test the metamaterial system. (b) Transmissions when vibration sources are excited at six different locations. (c) Correlation of the transmissions. (d) Vibration modes

of the metamaterial when excited at different locations at 695.3 Hz.

The signal set contains 20 broadband testing signals. The testing signals in the signal set are labeled as #1 ~ #20. The signals used in Figs. 4(a) – 4(d) are the testing signals with the labels. In the revised manuscript, we have highlighted the construction of the testing signals in the signal set in the main text and the Methods as follows:

“Twenty different testing signals with normalized energy are collected in a signal set (see Methods for the construction of the testing signals).” (Page 4, Left column, Line 4)

“For the identification tasks, we construct 20 testing signals as a signal set to verify the performance of the metamaterial system. Each signal is constructed by superposing randomized sine waves as $s(t) = \sum_{i=1}^{20} a_i \sin(2\pi f_i t + \varphi_i)$, where $t \in [0, 1]$, $a_i \in [0, 1]$, $f_i \in [200, 800]$, and $\varphi_i \in [0, 2\pi]$. Energy normalization is performed by dividing the root mean square of each signal to ensure that the testing signals have the same intensity. The waveforms and spectra of the 20 testing signals are provided in Supplementary Note 5.” (Page 7, Right column, Line 75)

Specifically, the waveforms of the testing signals are presented as follows:

Supplementary Figure 7. Waveforms and spectra of the twenty testing signals in the signal set.

Reviewer's Comment:

4. The supplementary information only has one paragraph, and the Method section only contains simple explanations on “Construction of the measurement matrix” and “Dimensionality compression of the measurement matrix”. As a general reader, the reviewer found difficult to reproduce the results demonstrated in the manuscript. The reviewer feels signal processing part may be more important than the conceptual part. Because the concept is not totally new. The technique part may be an important contribution, if the authors can provide detailed information for this judgement. Taking an example to demonstrate this in details may be a good strategy.

Authors' Response:

Thanks for the reviewer's constructive comments and suggestions. We firstly moved the part “Dimensionality compression of the measurement matrix” into the part “Construction of the measurement matrix in frequency domain” and added more details in the revised supplementary information. The revised part is specifically presented as follows:

“The measurement matrix contains the spatial vibration encoding information of the metamaterial and the content of the testing signals. Before using the metamaterial system for vibration identification, we need to know the *a priori* measurement matrix. The construction of the measurement matrix includes two steps: experimental calibration and dimensionality compression.

(i) Experimental calibration is currently an effective way to obtain the accurate transmissions due to the unavoidable manufacturing tolerance and the complex testing environments. Theoretically, the measurement matrix $\mathbf{M}_{p \times q}$ can be expressed by $M_q(\omega_p) = H_k(\omega_p) \cdot S_j(\omega_p)$, where p is the index of frequency, k is the index of the location, j is the index of the testing signal, $q = k \times j$ is the column number of measurement matrix, $H_k(\omega_p)$ is the transmission of the metamaterial system, and $S_j(\omega_p)$ is the spectrum of the testing signal. In our experiments, the measurement matrix is calibrated by successively playing the testing signals from different locations and calculating the spectra of the measured signals. More details can be found in Supplementary Note 6.

(ii) Dimensionality compression. The current calibrated measurement matrix $\mathbf{M}_{p \times q}$ contains redundant information as well as the noise. After the mixed vibration signal $\mathbf{y}_{p \times 1}$ (i.e. observation vector) is acquired, we respectively replicates the matrices $\mathbf{y}_{p \times 1}$ and $\mathbf{M}_{p \times q}$ by row for r times ($r > 1$ and $r \in \mathbb{Z}$) to increase the weight of the effective information before using dimensionality compression algorithms. The principal component analysis is used to compress the dimensionality of the measurement matrix. First, $\mathbf{M}_{rp \times q}$ and $\mathbf{y}_{rp \times 1}$ are assembled into matrix $\mathbf{A}_{rp \times (q+1)}$. Then, we perform zero-mean normalization on $\mathbf{A}_{rp \times (q+1)}$. Next, singular value decomposition is performed on $\mathbf{A}\mathbf{A}^T$ to obtain the left singular vector matrix $\mathbf{U}_{rp \times rp}$ and the singular value matrix \mathbf{D} . By selecting the first p' principal singular values of \mathbf{D} and transforming $\mathbf{U}_{rp \times rp}$ to $\mathbf{U}_{rp' \times p'}$, the compressed measurement matrix can be obtained by $\mathbf{M}_{p' \times q} = \mathbf{U}_{rp' \times p'}^T \mathbf{M}_{rp \times q}$. In this way, the principal component analysis can compress the measurement matrix to reduce the computational complexity while maintaining the reconstruction accuracy and further improving the robustness against noises.” (Page 7, Right column, Line 87)

Then, we added an example to demonstrate the signal processing part in details in the revised Supplementary Note 6 as the reviewer's suggestion. The results of each step of the signal processing are shown in Supplementary Fig. 8. Specifically, the revisions are presented as follows:

“... Specifically, we take an example to demonstrate the entire identification process in detail in Supplementary Note 4.” (Page 4, Left column, Line 22)

“In our experiments, the measurement matrix $\mathbf{M}_{p \times q}$ is experimentally calibrated by successively playing the 20 testing signals from six different locations and calculating the spectra of the measured signals (see Supplementary Fig. 8(a)). The operational frequency range is 200 ~ 800 Hz, the sampling frequency is 12.82 kHz, $p = 1534$, and $q = 120$. The observation vector $\mathbf{y}_{p \times 1}$ is obtained by playing testing signals from multiple sources at the same time. Then, we replicates the matrices $\mathbf{y}_{p \times 1}$ and $\mathbf{M}_{p \times q}$ by row (now, $p = 1 \times 10^4$) to increase the weight of the effective information as shown in Supplementary Fig. 8(b). Next, principal component analysis is used to compress the dimensionality of the measurement matrix. Here, the first 100 principal singular values are selected. The compressed \mathbf{M} and \mathbf{y} are presented in Supplementary Fig. 8(c).

The solution of $\mathbf{y} = \mathbf{M}\mathbf{x}$ can be obtained with an L1-norm minimization

$$\hat{\mathbf{x}} = \min \|\mathbf{x}\|_1, \quad \text{s.t. } \mathbf{M}\mathbf{x} = \mathbf{y}, \quad (\text{S4})$$

where $\hat{\mathbf{x}}$ is the estimation of the unknown \mathbf{x} , and $\|\mathbf{x}\|_1$ denotes the L1-norm $\sum_i |\mathbf{M}\mathbf{x}_i - \mathbf{y}_i|$ of \mathbf{x} . We use the two-step iterative shrinkage/thresholding algorithm to solve this inverse problem. The normalized absolute $\hat{\mathbf{x}}$ can be divided into six parts according to the number of locations as shown in Supplementary Fig. 8(d). It can be seen that the testing signal #11 is generated from Location 2, and the testing signal #5 is generated from Location 3, which agrees well with the truth. Finally, we visualize and present the reconstructed result in Supplementary Fig. 8(e).” (Supplementary Note 6)

Supplementary Figure 8. An example of the signal processing details of the vibration identification. (a) The calibrated measurement matrix \mathbf{M} and observed vector \mathbf{y} . (b) The replicated \mathbf{M} and \mathbf{y} . (c) The compressed \mathbf{M} and \mathbf{y} by using principal component analysis. (d) The details and (e) the visualization of the reconstructed results.

In the revised manuscript, we added new experiments to show the trajectory tracking of vibration events based on the identification method of impacts. For the impact identification and the additional experiments, we also included more details to strength the reproducibility of our approach.

The revisions are presented as follows:

“The convolution process of impact identification for the randomized resonant metamaterial system can be expressed as

$$y(t) = \int_0^t h(t-\tau)f(\tau)d\tau, \quad (S5)$$

where $y(t)$ is the observation signal, $h(t)$ is the impulse response function, and $f(t)$ is the impact function. The discrete form of Eq. S5 can be given by

$$y(n\Delta t) = \Delta t \sum_{i=1}^n [h((n-i+1)\Delta t) \cdot f(i\Delta t)], \quad (S6)$$

where Δt is the time interval. Eq. S6 can be written as the following matrix-vector form

$$\begin{bmatrix} y(\Delta t) \\ y(2\Delta t) \\ \vdots \\ y((n-1)\Delta t) \\ y(n\Delta t) \end{bmatrix} = \Delta t \begin{bmatrix} h(\Delta t) & 0 & \cdots & 0 & 0 \\ h(2\Delta t) & h(\Delta t) & \cdots & 0 & 0 \\ \vdots & \vdots & \ddots & \vdots & \vdots \\ h((n-1)\Delta t) & h((n-2)\Delta t) & \cdots & h(\Delta t) & 0 \\ h(n\Delta t) & h((n-1)\Delta t) & \cdots & h(2\Delta t) & h(\Delta t) \end{bmatrix} \begin{bmatrix} f(\Delta t) \\ f(2\Delta t) \\ \vdots \\ f((n-1)\Delta t) \\ f(n\Delta t) \end{bmatrix}. \quad (S7)$$

Therefore, the measurement matrix for impact identification can be constructed by continuously delaying the directional impulse responses of the metamaterial system with time interval Δt in a time window. In our experiments, the impulse responses of the metamaterial system are directly obtained by applying impulses to different locations. Then, time alignment and amplitude normalization are performed to obtain the calibrated impulse responses $h^k(t)$, where k is the location index. The measurement matrix \mathbf{M} for impact identification can be constructed as $\mathbf{M} = [\mathbf{H}^1, \mathbf{H}^2, \dots, \mathbf{H}^6]$, where

$$\mathbf{H}^k = \begin{bmatrix} h^k(\Delta t) & 0 & \cdots & 0 & 0 \\ h^k(2\Delta t) & h^k(\Delta t) & \cdots & 0 & 0 \\ \vdots & \vdots & \ddots & \vdots & \vdots \\ h^k((n-1)\Delta t) & h^k((n-2)\Delta t) & \cdots & h^k(\Delta t) & 0 \\ h^k(n\Delta t) & h^k((n-1)\Delta t) & \cdots & h^k(2\Delta t) & h^k(\Delta t) \end{bmatrix}, \quad k = 1, 2, \dots, 6. \quad (S8)$$

(Supplementary Note 9)

“The waveform presented in Fig. 5(a) is the whole measured signal in the trajectory tracking. We continuously intercept the measured signal according to each vibration event as shown in Supplementary Fig. 13(a). Here we take an example to show the identification performance, where the 7th fragment of the measured signal is used to be the observation vector. Supplementary Fig. 13(b) shows the normalized reconstruction result of the selected vibration event. The occurrence location and time of the reconstructed impact are in good agreement with the actual one. We calculate the maximum value of the reconstructed vector segment corresponding to each location in Supplementary Fig. 13(b). The maximum values are visualized in Supplementary Fig. 13(c). It can be seen that an impact is applied to the probe C. ...” (Supplementary Note 10)

Supplementary Figure 13. Details of the reconstruction for trajectory tracking. (a) The measured signal from the single sensor. (b) The normalized reconstruction results of the 7th vibration event, and (c) the visualization of the maximum values in the reconstructed vector segments.

Reviewer’s Comment:

5. Is specially designed disorder better than the randomized metamaterial? Firstly, the specially designed disorder can maximize the orthogonality with minimum quantities of metamaterial unit cells. Second, the specially designed disorder can be analyzed, which could avoid pre-experimental calibrations.

Authors’ Response:

Thanks for the reviewer’s constructive comments and suggestions. First, we agree with the reviewer that the special design approach can maximize the uncorrelation (orthogonality) of the vibration transmission property and reduce the number of unit cells. To verify this point, we have investigated the influence of the number of the locally resonant unit cells on the uncorrelation by using the revised simplified model. We have included the discussion in the revised manuscript, and the statements are specifically presented as follows:

“The supercell network contains 19 locally resonant unit cells. To study the effect of reducing the number of the locally resonant unit cells on the vibration transmission property, we randomly replace the effective masses m_n^{eff} with the matrix masses M in each supercell network, and calculate the μ_{Ave} of the entire system (Supplementary Fig. 6(a)). Each random replacement process is conducted 10 times, and the average, maximum, minimum values of the μ_{Ave} obtained from 10 times of calculations are shown in Supplementary Fig. 6(b). It can be seen that the average of the μ_{Ave} is enlarged by reducing the number of locally resonant unit cells, but the μ_{Ave} in some special cases is smaller than that with 19 unit cells. This means that by further optimizing the design, the maximum uncorrelation of the transmissions can be achieved with minimum quantities of unit cells.” (Supplementary Note 4)

Supplementary Figure 6. Effects of the number of the locally resonant unit cells on the μ_{Ave} . (a) Schematic of randomly replacing four effective masses m_{eff} with four mass M in each supercell network. (b) The relationship between the μ_{Ave} and the number of the locally resonant unit cells in each supercell network. Here, each point denotes the average of the μ_{Ave} in 10 calculations, and the error bar denotes the maximum and minimum values of the μ_{Ave} in 10 calculations.

This manuscript mainly focuses on the proof-of-concept of the spatial vibration information encoding and the single-sensor identification of vibrations. We have provided the metric μ_{Ave} to quantify the uncorrelation of the vibration transmission property, which can be used as an objective function for the optimization design in further work. By using topological optimization and machine learning methods, optimal disorder structures can be designed and analyzed, but it is beyond the subject of this manuscript.

Second, we agree that the specially designed disorder can be easily analyzed, but pre-experimental calibrations is still indispensable currently due to the indeterminacy of material parameters and the manufacturing tolerance. Although precision machining can be used to reduce the error between the sample and the design, pre-experimental calibration is currently an effective way to obtain accurate transmissions to construct the measurement matrix under the complex practical environments. How to bridge the gap between the design and the experiment to avoid the calibrations remains a challenge that needs to be addressed in the further work.

The discussion included in the revised manuscript is specifically presented as follows:

“However, there are also limitations for the proposed approach. First, although the proposed metamaterial system has achieved an effective spatial vibration encoding, the randomized design strategy is not the optimal one that can maximize the uncorrelation while minimizing the number of unit cells. This limitation can be overcome by using topological optimization^{34,46} or machine learning methods⁴⁷ to achieve the on-demand design of the disordered structures in the further work. Moreover, the proposed approach needs the pre-experimental calibrations because the actual sample in experiments might not match the designed one due to the unavoidable manufacturing tolerance. This issue is hoped to be solved by utilizing artificial intelligence algorithms based on the computational and experimental data^{48,49}.

34. Estakhri, N. M., Edwards, B. & Engheta, N. Inverse-designed metastructures that solve equations. *Science* **363**, 1333-1338 (2019).

46. Dong, H.-W., Zhao, S.-D., Wang, Y.-S. & Zhang, C. Topology optimization of anisotropic broadband double-negative elastic metamaterials. *J. Mech. Phys. Solids* **105**, 54-80 (2017).

47. Ma, W., Cheng, F. & Liu, Y. Deep-learning-enabled on-demand design of chiral metamaterials. *ACS Nano* **12**, 6326-6334 (2018).
48. Wang, F., Wang, H., Wang, H., Li, G. & Situ, G. Learning from simulation: An end-to-end deep-learning approach for computational ghost imaging. *Opt. Express* **27**, 25560-25572 (2019).
49. Jha, D. *et al.* Enhancing materials property prediction by leveraging computational and experimental data using deep transfer learning. *Nat. Commun.* **10**, 1-12 (2019).” (Page 6, Left column, Line 22)

Reviewer’s Comment:

6. What are the difficulties to increase the source locations from six to larger quantities? Is this physically realizable?

Authors’ Response:

Thanks for the reviewer’s constructive comments. The difficulty in increasing the number of source locations is whether the correlation of the spatial vibration transmissions is small enough. In our revised manuscript, we have added new experiments to demonstrate the application of the proposed metamaterial system for trajectory tracking of vibration events, where the source locations are increase to eight and twelve. We calculate the μ_{Ave} of the directional impulse responses of the two cases (with eight and twelve locations). The values are approximately equal to 0.10. The additional experiments and results are specifically presented as follows:

“To exhibit the broad application prospects, we demonstrate a use of the proposed metamaterial system for trajectory tracking of vibration events. Here, eight probes (denoted by ‘A’ to ‘H’) are fixed around the metamaterial as shown in Fig. 5(a). We tap the probes with the finger in sequence along the illustrated trajectory ‘A-E-F-B-H-D-C-G-A’, and the vibration signals are measured as shown in Fig. 5(a) by the single sensor. We can clearly see the tracked trajectory according to the identification of vibration events. Figure 5(b) shows the reconstructed results of each vibration event as well as the trajectory pattern, where the occurrence locations and time agree well with the truth (see Supplementary Note 10 for details). This trajectory tracking process is dynamically presented in Supplementary Video 1. We also study the tracking process of the trajectories ‘SJTU’ (i.e. the abbreviation of Shanghai Jiao Tong University). The tracked trajectories are well exhibited in Fig. 5(c) (see Supplementary Fig. 14 for the details). Figure 5(d) shows the successfully tracked trajectory ‘Vase’ with 12 probes. The experimental details and reconstruction results are presented in Supplementary Fig. 15 and Supplementary Video 2. The results above demonstrate that the proposed metamaterial system can be used as a new type of human-machine interface for instruction, communication, and encryption without complex hardware and high power consumption. Moreover, it also has potential application prospects in fields such as robot tactile sensing and collision tracking.” (Page 5, Left column, Line 23)

Fig. 5. Trajectory tracking of vibration events. (a) Experimental setup for trajectory tracking with eight probes. The probes are tapped along the trajectory ‘A-E-F-B-H-D-C-G-A’, and the vibration signals are measured by the single sensor. The trajectory is successfully tracked according to the identification of vibration events. (b) Reconstruction results of vibration events. The dynamic process can be found in Supplementary Video 1. (c) The tracked trajectories of ‘SJTU’. (d) The tracked trajectory ‘Vase’ with 12 probes. The dynamic process can be found in Supplementary Video 2.

“The waveform presented in Fig. 5(a) is the whole measured signal in the trajectory tracking. We continuously intercept the measured signal according to each vibration event as shown in Supplementary Fig. 13(a). Here we take an example to show the identification performance, where the 7th fragment of the measured signal is used to be the observation vector. Supplementary Fig. 13(b) shows the normalized reconstruction result of the selected vibration event. The occurrence location and time of the reconstructed impact are in good agreement with the actual one. We calculate the maximum value of the reconstructed vector segment corresponding to each location in Supplementary Fig. 13(b). The maximum values are visualized in Supplementary Fig. 13(c). It can be seen that an impact is applied to the probe C. Furthermore, we use a similar method to reconstruct the trajectories ‘SJTU’ as shown in Supplementary Fig. 14.

Supplementary Fig. 15 shows the reconstruction details of the trajectory ‘Vase’, where the probes are increased to 12. It can be seen that the occurrence locations and time of the vibration events can still be successfully identified. This tracking process is dynamically shown in Supplementary Video 2. The results above demonstrate that the proposed device can be used to track complex trajectories, which has potential application prospects in fields such as human-machine interface and collision tracking.” (Supplementary Note 10)

Supplementary Figure 13. Details of the reconstruction for trajectory tracking. (a) The measured signal from the single sensor. (b) The normalized reconstruction results of the 7th vibration event, and (c) the visualization of the maximum values in the reconstructed vector segments.

Supplementary Figure 14. Tracking process of the trajectories 'SJTU'. (a)-(d) The tracked trajectories, measured signals, and reconstruction results of 'S', 'J', 'T', and 'U', respectively.

Supplementary Figure 15. Tracking process of the trajectory ‘Vase’. (a) Experimental setup. (b) The tracked trajectory. (c) The measured signal from the single sensor. (d) Reconstruction results of each vibration event.

Reviewer’s Comment:

7. In Fig. 2(a), the simplified model contains dampers. What are the damping effects on the identification?

Authors’ Response:

Thanks for the reviewer’s constructive comments. Because the absolute average cross-correlation μ_{Ave} is a metric to evaluate the performance of the identification [1], we can study the damping effects on identification by studying the damping effects on μ_{Ave} . We have modified the simplified model to a general metamaterial model to more accurately investigate the vibration transmission property. The relationship between μ_{Ave} and the damping coefficients is investigated based on the modified general metamaterial model. The results are shown in Supplementary Fig. 5(a). It can be seen that the damping of the unit cell c_n has a great influence on μ_{Ave} , while the damping of the matrix c_0 has a relatively small effect on μ_{Ave} . Therefore, materials with appropriate damping should be selected to achieve the optimum identification performance. We have added the related discussions in the revised manuscript, and the discussions are specifically presented as follows:

“We also investigate the effects of the model parameters on the vibration transmission property of the general metamaterial model. The relationship between the μ_{Ave} and the parameters, including c_n , c_0 , k_0 , and the number of the locally resonant unit cells, can be found in Supplementary Note 4. The results show that the stiffness of the matrix k_0 , the damping of the unit cells c_n , and the number of the locally resonant unit cells have a great influence on the uncorrelation of the vibration transmission, which provides a guidance for the optimization model design.” (Page 2, Right column, Line 47)

“The absolute average cross-correlation μ_{Ave} is a metric to evaluate the uncorrelation of the vi-

bration transmission, which is highly related to the identification performance. To study the parameter effects on the transmission property of the general metamaterial model, we calculated the μ_{Ave} under different c_n (damping of the unit cell), c_0 (damping of the matrix) and k_0 (stiffness of the matrix). The results are shown in Supplementary Fig. 5. It can be seen that c_n and k_0 have a great influence on μ_{Ave} , while c_0 has a relatively small effect on μ_{Ave} . Therefore, materials with appropriate damping and stiffness should be selected to achieve the optimum identification performance.” (Supplementary Note 4)

Supplementary Fig. 5. The relationship between the μ_{Ave} and the dynamic parameters of the general metamaterial model. (a) Effects of the damping c_n and c_0 when $k_0 = 1.9 \times 10^5$ N/m. (b) Effects of the matrix stiffness k_0 when $c_0 = 0.01$ N·s/m and $c_n = 0.25$ N·s/m.

- [1] Xie, Y. *et al.* Single-sensor multispeaker listening with acoustic metamaterials. *Proc. Natl. Acad. Sci. U.S.A.* **112**, 10595-10598 (2015).

We would like to thank the reviewer for the positive and constructive review on our work again and hope these revisions are satisfactory.

Reviewers' comments:

Reviewer #1 (Remarks to the Author):

The paper is well written and the comments of the other two referees have been properly answered so that the paper is now of excellent quality.

However, the authors have not answered properly to my complaints. I agree with them that the applications of vibrations are different than those of acoustics and electromagnetics, but what I mean is that the concept is not new, since it has been widely demonstrated before, therefore I consider that the work is not innovative enough for Nature Communications.

Reviewer #2 (Remarks to the Author):

Authors answered properly to reviewer's questions.
New interesting applications and future works can be envisioned.
I recommend the paper for publication.

Reviewer #3 (Remarks to the Author):

The reviewer thanks authors for their hard work in revising the manuscript, which contains a significant shape change compared with the original submission and a very long rebuttal letter. After carefully reading all the submissions, the reviewer thinks the authors have mostly answered my questions, added necessary details for their vibration identifications and made corresponding changes. A new figure (Fig. 5) has been added into the main text. The reviewer understood the authors may want to make the results more attractive, however this is unnecessary from the result reporting aspect. For me, Figure 5 serves the same purpose as Figs. 4e – 4g to demonstrate impact identifications, which is therefore not an important result to put into the main text. The authors can make their own decision to keep it or remove it. The reviewer thinks the main contribution of this manuscript is to utilize the concept of the compressive sensing in identification of vibration sources, which could be a novelty for publication in Nat Comm, according to my own understanding and criterion. Besides these, the reviewer also has two other comments for the final version:

1. Based on the SI, the reviewer found the vibration identification needs, at least, some pre-knowledge of the excitation signals of vibration sources. However, this has not been properly or explicitly told in the main text. For example, the authors discussed some drawbacks, but ignored this important point for some reasons.
2. Regarding the metamaterial modeling part, which has now become an important section in the revised version, some important and highly-cited references in this field have however been missed by the authors, i.e.

[1] HH Huang, CT Sun, GL Huang, On the negative effective mass density in acoustic metamaterials, International Journal of Engineering Science 47 (4), 610-617, 2009.

[2] R Zhu, XN Liu, GK Hu, CT Sun, GL Huang, A chiral elastic metamaterial beam for broadband vibration suppression, Journal of Sound and Vibration 333 (10), 2759-2773, 2014.

[3] Y.Y. Chen, G.K. Hu and G.L. Huang (2017) "A Hybrid Elastic Metamaterial with Negative Mass Density and Extremely Tunable Modulus", Journal of Mechanics and Physics Solids, 105: 179-198.

Randomized resonant metamaterials for single-sensor identification of elastic vibrations

Tianxi Jiang, Chong Li, Qingbo He, and Zhi-Ke Peng

Reviewer's Comment:

The paper is well written and the comments of the other two referees have been properly answered so that the paper is now of excellent quality.

Authors' Response:

We thank the reviewer very much for the positive comments on the quality of our revised manuscript.

Reviewer's Comment:

However, the authors have not answered properly to my complaints. I agree with them that the applications of vibrations are different than those of acoustics and electromagnetics, but what I mean is that the concept is not new, since it has been widely demonstrated before, therefore I consider that the work is not innovative enough for Nature Communications.

Authors' Response:

We appreciate the reviewer's comments. This gives us a good chance to further deeply think about the main contributions of our work compared to those of acoustics and electromagnetics. In the revisions, we have further strengthened the significant theoretical contribution of our work. In the following, we summarize the significance and broad interests of our work.

We believe that our work can open up avenues not only in vibration sensing fields, but also in elastic metamaterial fields, thereby attracting broad interests of the readership. We first, to our knowledge, use the elastic metamaterial with randomized local resonators to encode and identify spatial low-frequency vibrations. The proposed approach provides a new perspective for designing the vibration transmissions to make the metamaterial has the special function of information coding and transmission. This work opens up a wide direction of vibration transmission encoding with plentiful potential applications. This manuscript provides one of the applications on the simpler vibration sensing with a single sensor, which demonstrates the possible time-dependent space coding ability to realize a new kind of human-machine interaction of pattern information. Besides the application demonstrated in this manuscript, there are also other plentiful application potentials, such as in fields of signal filtering, information communication, etc.

In addition, elastic metamaterials have been an active research area in the last decade, which has caused revolutions in the manipulation of elastic waves and the design of new devices. However, to our knowledge, there have been no reports about the use of elastic metamaterials for vibration transmission encoding and identification. We believe that our study can promote the development of the elastic

metamaterial field.

In the revised manuscript, we further summarize and refine a novel concept of randomly coupled resonator dynamics to describe the design idea of the proposed metamaterial. The proposed concept has two connotations: (a) The proposed randomly coupled resonator system consists of multiple random local resonators; (b) The coupling relationship of resonators are spatially disordered. From the dynamics theory of the randomly coupled resonator system, we can find that the high uncorrelation of vibration transmissions are determined by the effective mass matrix and the coupling matrix. The effective mass matrix contains the frequency-dependent effective masses of the random resonators, and the effective mass can be negative near the resonant frequency of the local resonator. The coupling matrix reflects the spatial coupling relationship of the random resonators. Therefore, the disordered coupling of the effective masses is the physical basis for effective spatial vibrations encoding. We use this dynamical model to design an actual metamaterial system, and we first achieve the compressive sensing in identification of vibration sources with a single sensor based on the proposed metamaterial. The proposed theoretical concept is a dynamics method that incorporates physical mechanism. This concept is important and universal for analyzing the propagation of elastic vibrations in such complex systems. It can also provide a basis for designing various types of metamaterials in future works, and open up attractive avenues to simpler sensing devices with low power cost for many other physical information.

The major points discussed above have been included in the revised manuscript, and are specifically presented as follows:

“In this work, we propose a randomized resonant metamaterial with randomly coupled local resonators for single-sensor identification of elastic vibrations. This metamaterial is designed by developing the theory of randomly coupled resonator dynamics. The metamaterial is proved to be capable of producing highly uncorrelated transmissions for different spatial vibrations due to the disordered coupling of random effective masses. ...” (Page 1, Line 63)

“To achieve the highly uncorrelated transmissions for spatial vibration information encoding, we propose a concept of randomly coupled resonator dynamics and develop the corresponding system. The proposed system consists of multiple different coupling networks. Each coupling network is composed of N local resonators connected by springs k_0 and dampers c_0 (Fig. 1(a)). k_n and c_n are the stiffness and the damping coefficients of the n th resonator, respectively. x_n and y_n are displacements of matrix M and mass m . We derive the randomly coupled resonator dynamics (see Methods for details). In the dynamics, the local resonator is simplified to be an effective mass as below

$$m_n^{\text{eff}} = M + \frac{k_n^d m}{k_n^d - \omega^2 m}, \quad (1)$$

where $k_n^d = k_n + i\omega c_n$, $k_n = m\omega_n^2$, and ω_n is the resonant frequency of the n th local resonator⁴⁰...

[Methods]

Analytical derivation of the randomly coupled resonator dynamics

The dynamical equation of the n th resonator can be expressed as

$$\begin{cases} M\ddot{x}_n + c_0 \left(\sum_{i=1}^N \psi_{ni} \dot{x}_i \right) + k_0 \left(\sum_{i=1}^N \psi_{ni} x_i \right) - c_n (\dot{y}_n - \dot{x}_n) - k_n (y_n - x_n) = F_n, \\ m\ddot{y}_n + c_n (\dot{y}_n - \dot{x}_n) + k_n (y_n - x_n) = 0, \end{cases} \quad (2)$$

where

$$\psi_{ni} = \begin{cases} -1, & i \text{ couples with } n, \\ \phi_n, & i = n, \\ 0, & \text{others,} \end{cases} \quad (3)$$

ϕ_n is equal to the number of the resonators coupling with the n th resonator. Eq. 2 can be derived as

$$\left(M + \frac{k_n^d m}{k_n^d - \omega^2 m} \right) \ddot{x}_n + (k_0 + i\omega c_0) \left(\sum_{i=1}^N \psi_{ni} x_i \right) = F_n \quad (4)$$

The expression of the effective mass (Eq. (1)) can thus be obtained. ...” (Page 1, Line 83)

“... From the randomly coupled resonator dynamics, we can find that this complex vibration propagation is determined by the effective mass matrix and the coupling matrix of the coupling system. These two matrices reflect the disordered coupling relationship of random effective masses. The disordered coupling of the effective masses leads to the high uncorrelation of vibration transmissions, which is the physical basis for effective spatial vibration encoding.” (Page 2, Line 45)

“The results above demonstrate the time-dependent space coding ability of the proposed metamaterial system, which can create a new type of human-machine interface for instruction, communication, and encryption without complex hardware and high power consumption.” (Page 5, Line 56)

“The proposed randomly coupled resonator system with disordered effective masses provides the theoretical basis for designing the randomized resonant metamaterial and is easy to be extended. The theoretical concept is a dynamics method that incorporates physical mechanism. The physical mechanism of spatial vibration encoding is that the disordered coupling of effective masses leads to uncorrelated vibration transmissions.” (Page 6, Line 11)

“This work opens up avenues of vibration transmission encoding with metamaterials. We envision that the proposed metamaterial can integrate with numerous intelligent devices, platforms, and structures (e.g., wearable devices, quadrotor drones, and airplane wings) so that it can be used in broad

fields such as human-machine interaction, health care monitoring, industrial field detection, information processing and communication.” (Page 6, Line 61)

Furthermore, the physical mechanism of the proposed model for compressive sensing is novel and different from the previous works (Refs. [1] – [3]) mentioned by the reviewer in the first round of review. In our study, the proposed metamaterial is composed of randomized local resonators, which is particularly suitable for manipulating low-frequency broadband vibrations. The physical mechanism of the proposed metamaterial is based on the disordered coupling of the effective masses. Because of this, the elastic vibration propagation in the metamaterial system is highly complex, which leads to the high uncorrelation of the spatial vibration transmissions. Moreover, the modulation of the vibration signals is the synergy of the resonance and anti-resonance of the local resonators. For a single unit cell, the vibration transmission can be enhanced near the resonance frequency and attenuated at the resonance frequency [4]. For the entire randomized resonant metamaterial in our study, the vibration signals can thus be enhanced to some extent in certain frequency bands, which is superior to the band-stop filtering mechanism [2] and benefit the vibration identification. We have added the related discussions in the revised manuscript as follows:

“The physical mechanism of spatial vibration encoding is that the disordered coupling of effective masses leads to uncorrelated vibration transmissions. In addition, the vibration transmission property of the metamaterial is the synergy of the resonance and anti-resonance of the local resonators. Vibration signals can thus be enhanced to some extent in certain frequency bands, which is superior to the band-stop filtering mechanism¹⁷ and benefit the vibration identification.

17. Xie, Y., Tsai, T. H., Konneker, A., Popa, B. I., Brady, D. J., & Cummer, S. A. (2015). Single-sensor multispeaker listening with acoustic metamaterials. *Proceedings of the National Academy of Sciences*, 112(34), 10595-10598.” (Page 6, Line 15)

We mentioned that our work demonstrates the possible time-dependent space coding ability for the elastic vibrations. Specifically, our study can not only identify the source locations, but also can identify the occurrence time of impact signals by using the temporal sparsity, which is also an innovation compared to other works. The measurement matrix for impact identification is constructed by using the impulse responses of the metamaterial. We successfully verify the feasibility of the proposed metamaterial system for impact identification. Accordingly, we demonstrate a novel application example of the proposed metamaterial system to track the trajectory of vibration events. We envision that our approach has potential application prospects in areas such as smart devices, Internet of Things, and human-machine interaction.

Based on the above points, our study can attract broad interests in the following aspects. First, this work opens up a wide direction of vibration transmission encoding with metamaterials on plentiful

potential applications, such as the human-machine interaction, signal filtering, information communication, etc. Second, the proposed randomly coupled resonator dynamics provides a powerful tool for analyzing the transmission properties of spatial vibrations. This dynamics theory incorporates the physical mechanism of the metamaterial, which can be used for designing various attractive metamaterials. Third, the designed metamaterial achieves the single-sensor vibration identification by combining with compressive sensing framework, which can inspire other artificial structures for the ideas of vibration sensing and information processing. The application prospects demonstrated in the manuscript can open up ideas for developing new type of smart devices.

Finally, we think that the novelty of concept in our work should be considered in conjunction with the specific research field. We agree with the reviewer that the concept of “compressive sensing and random metamaterials” is a broad concept and has been demonstrated in acoustics and electromagnetics. The core of this concept is that the measurement matrix of compressive sensing can be physically realized by metamaterials. Thus, the key issue is to develop an eligible metamaterial to spatially encode different physical fields. In different research fields, the structure design ideas and signal processing methods are different. For example, in ultrasonics, the working frequency for compressive ultrasound imaging in Ref. [1] (mentioned by the reviewer in the first round of review) is a single ultrasonic frequency. Besides, the wavelength of the ultrasound is so small that the random scattering mask can be used to encoding spatial ultrasound field for compressive 3D imaging. In audio acoustics, the device in Ref. [2] is based on the acoustic resonance absorption of the Helmholtz resonators. This mechanism is essentially a type of band-stop filtering, which may reduce the energy of the received signals. In electromagnetics, the device in Ref. [3] operates under an impedance matching scheme to modulate the electromagnetic waves, which is also not applicable to the field of elastic vibrations. For elastic vibrations considered in our work, signals are usually in low-frequency ranges (less than 1000 Hz) and broadband. Our study aims to fill the gap of knowledge in designing the metamaterial that can encode low-frequency broadband elastic vibrations for compressive identification, and opens up a wide direction of vibration transmission encoding with metamaterials. This study is of great significance in the fields of vibration sensing/communication and elastic metamaterials, and can appeal to the broad interest of the readership of *Nature Communications*.

We would like to thank the reviewer for the careful review again, and hope these revisions are satisfactory.

Reference:

- [1] Kruizinga, P., van der Meulen, P., Fedjajevs, A., Mastik, F., Springeling, G., de Jong, N., ... & Leus, G. (2017). Compressive 3D ultrasound imaging using a single sensor. *Science advances*, 3(12), e1701423.
- [2] Xie, Y., Tsai, T. H., Konneker, A., Popa, B. I., Brady, D. J., & Cummer, S. A. (2015). Single-sensor multispeaker listening with acoustic metamaterials. *Proceedings of the National Academy of Sciences*, 112(34), 10595-10598.

- [3] Watts, C. M., Shrekenhamer, D., Montoya, J., Lipworth, G., Hunt, J., Sleasman, T., ... & Padilla, W. J. (2014). Terahertz compressive imaging with metamaterial spatial light modulators. *Nature Photonics*, 8(8), 605.
- [4] Jiang, T., & He, Q. (2017). Dual-directionally tunable metamaterial for low-frequency vibration isolation. *Applied Physics Letters*, 110(2), 021907.

Response to Comments of Reviewer #2

Randomized resonant metamaterials for single-sensor identification of elastic vibrations

Tianxi Jiang, Chong Li, Qingbo He, and Zhi-Ke Peng

Reviewer's Comment:

Authors answered properly to reviewer's questions.

New interesting applications and future works can be envisioned.

I recommend the paper for publication.

Authors' Response:

We appreciate the reviewer very much for the positive comments on the revised manuscript.

Reviewer's Comment:

The reviewer thanks authors for their hard work in revising the manuscript, which contains a significant shape change compared with the original submission and a very long rebuttal letter. After carefully reading all the submissions, the reviewer thinks the authors have mostly answered my questions, added necessary details for their vibration identifications and made corresponding changes.

Authors' Response:

We thank the reviewer very much for the positive comments on the first round of our revisions, and the constructive comments that helped us further improve the quality of this manuscript.

Reviewer's Comment:

A new figure (Fig. 5) has been added into the main text. The reviewer understood the authors may want to make the results more attractive, however this is unnecessary from the result reporting aspect. For me, Figure 5 serves the same purpose as Figs. 4e – 4g to demonstrate impact identifications, which is therefore not an important result to put into the main text. The authors can make their own decision to keep it or remove it.

Authors' Response:

Thanks for the reviewer's constructive comments. The new Fig. 5 in the revised manuscript is to demonstrate the usefulness of our approach. Although the results reported in Fig. 5 appear similar to Figs. 4e – 4g, there are some differences as listed as follows:

(1) **The excitation locations are different.** In Figs. 4e – 4g, the impacts are directly applied to the boundaries of the metamaterial. For Fig. 5, some excitations are not directly applied to the boundaries of the metamaterial, but applied to the probes around the metamaterial. The probes are fixed on a plate where the metamaterial is embedded. The following Fig. I shows the difference for these two cases. This will bring benefits for the case of Fig. 5, for example, the number of the source locations can be larger than the case of Fig. 4e if the spatial transmissions are uncorrelated enough. For these two cases in the manuscript, the number of the potential source locations in Fig. 5 (8 and 12 source locations) is indeed larger than that in Figs. 4e – 4g (6 source locations).

Fig. I. Excitation locations in Fig. 4e and Fig. 5a.

(2) **The verification aims are different.** Figs. 4e – 4g verify the feasibility of impact identifications in a general sense. However, Fig. 5 shows the application prospects of the proposed metamaterial system in areas such as time-dependent vibration event trajectory tracking for human-machine interaction (e.g. for instruction, communication, and encryption). The results in Fig. 5 and two Supplementary Videos clearly visualize the locations and trajectory tracking of the vibration events. We hope that Fig. 5 can inspire more interesting works based on our approach in areas such as robot tactile sensing and collision tracking.

For the above reasons, we choose to keep Fig. 5 as an important result to demonstrate the potential applications of our proposed metamaterial system. Accordingly, we made some revisions of the related statements as follows:

“The proposed metamaterial system can also be used to identify impacts as shown in Fig. 4(e). ... The results above verify the feasibility of the proposed metamaterial system for impact identification.

To demonstrate the broad application prospects of our approach, we use the proposed metamaterial system to track the trajectory of multiple vibration events. Here, eight probes (denoted by ‘A’ to ‘H’) are fixed on a plate where the metamaterial is embedded as shown in Fig. 5(a). ... This trajectory tracking process is dynamically visualized in Supplementary Video 1. ...” (Page 5, Line 33)

Reviewer’s Comment:

The reviewer thinks the main contribution of this manuscript is to utilize the concept of the compressive sensing in identification of vibration sources, which could be a novelty for publication in Nat Comm, according to my own understanding and criterion. Besides these, the reviewer also has two other comments for the final version:

Authors’ Response:

We thank the reviewer very much for the positive evaluation on our work, and the constructive

comments. All the comments and suggestions have been carefully considered in the revisions and the responses are provided as follows in detail.

Reviewer's Comment:

1. Based on the SI, the reviewer found the vibration identification needs, at least, some pre-knowledge of the excitation signals of vibration sources. However, this has not been properly or explicitly told in the main text. For example, the authors discussed some drawbacks, but ignored this important point for some reasons.

Authors' Response:

We thank the reviewer for pointing out our negligence. We have highlighted this point in the main text of the revised manuscript as follows:

“In this way, the sensing system can be expressed as $y = \mathbf{M}\mathbf{x}$, where y is the vector form of the measured data from the single sensor (i.e., observation vector), x is the object vector containing the information of sources, and \mathbf{M} is the measurement matrix determined by the encoding property of the metamaterial and the pre-knowledge of vibration excitations.” (Page 3, Line 11)

“Because the identification needs the *a priori* knowledge of the measurement matrix, a calibration process is experimentally performed in advance to construct the measurement matrix \mathbf{M} by calculating the spectra of the measured signals (see Methods).” (Page 4, Line 18)

“Moreover, the proposed approach needs the pre-experimental calibrations because the actual sample in experiments might not match the designed one due to the unavoidable manufacturing tolerance. The vibration identification is partly dependent on some pre-knowledge of vibration excitations, which is another reason for pre-experimental calibrations. This issue is hoped to be solved by utilizing artificial intelligence algorithms based on the computational and experimental data^{51,52}.” (Page 6, Line 51)

Reviewer's Comment:

2. Regarding the metamaterial modeling part, which has now become an important section in the revised version, some important and highly-cited references in this field have however been missed by the authors, i.e.

[1] HH Huang, CT Sun, GL Huang, On the negative effective mass density in acoustic metamaterials, International Journal of Engineering Science 47 (4), 610-617, 2009.

[2] R Zhu, XN Liu, GK Hu, CT Sun, GL Huang, A chiral elastic metamaterial beam for broadband vibration suppression, Journal of Sound and Vibration 333 (10), 2759-2773, 2014.

[3] Y.Y. Chen, G.K. Hu and G.L. Huang (2017) “A Hybrid Elastic Metamaterial with Negative Mass Density and Extremely Tunable Modulus”, Journal of Mechanics and Physics Solids, 105: 179-198.

Authors' Response:

We thank the reviewer for the constructive comment. We have added the important references in the manuscript as shown as follows:

“We derive the randomly coupled resonator dynamics (see Methods for details). In the dynamics, the local resonator is simplified to be an effective mass as below

$$m_n^{\text{eff}} = M + \frac{k_n^{\text{d}} m}{k_n^{\text{d}} - \omega^2 m}, \quad (1)$$

where $k_n^{\text{d}} = k_n + i\omega c_n$, $k_n = m\omega_n^2$, and ω_n is the resonant frequency of the n th local resonator⁴⁰.

40. Huang, H., Sun, C. & Huang, G. On the negative effective mass density in acoustic metamaterials. *Int. J. Eng. Sci.* **47**, 610-617 (2009).” (Page 1, Line 91)

“For the OSN, the transmission has a wide attenuation region (gray shaded region in Fig. 1(b)) due to the presence of the local resonance band gap⁴¹.

41. Zhu, R., Liu, X., Hu, G., Sun, C. & Huang, G. A chiral elastic metamaterial beam for broadband vibration suppression. *J. Sound Vib.* **333**, 2759-2773 (2014).” (Page 2, Line 5)

“We also calculate the frequency-dependent effective mass density⁴⁵. The out-of-plane band gap between 304.0 and 493.7 Hz is consistent with the frequency band of the negative effective mass density.

45. Chen, Y., Hu, G. & Huang, G. A hybrid elastic metamaterial with negative mass density and tunable bending stiffness. *J. Mech. Phys. Solids* **105**, 179-198 (2017).” (Page 3, Line 33)

We would like to thank the reviewer for the positive and constructive review on our work again and hope these revisions are satisfactory.

REVIEWERS' COMMENTS:

Reviewer #3 (Remarks to the Author):

The authors answer my questions properly

Reviewers' comments:

Reviewer #3 (Remarks to the Author):

The authors answer my questions properly.

Authors' Response:

We thank the reviewer for the positive comment on our revised manuscript.